# DeepRRTime:
# Robust Time-series Forecasting with a Regularized INR Basis

**Chandramouli Sastry**[†]    *chandramouli.sastry@gmail.com*
*Dalhousie University*
*Vector Institute*
*RBC Borealis*

**Mahdi Gilany**[†]    *mahdi.gilany@queensu.ca*
*Queen's University*
*RBC Borealis*

**Kry Yik-Chau Lui**    *yikchau.y.lui@borealisai.com*
*RBC Borealis*

**Martin Magill**    *martin.magill@borealisai.com*
*RBC Borealis*

**Alexander Pashevich**    *alexander.pashevich@borealisai.com*
*RBC Borealis*

**Reviewed on OpenReview:** *https://openreview.net/forum?id=uDRzORdPT7*

## Abstract

This work presents a simple, inexpensive, theoretically motivated regularization term to enhance the robustness of deep time-index models for time-series forecasting. Recently, Deep-Time demonstrated that this class of models can rival state-of-the-art deep historical-value models on the long time-series forecasting (LTSF) benchmarks. The DeepTime framework comprises two key components: (1) a time-indexed basis parameterized as an implicit neural representation (INR), and (2) a meta-learning formulation that fits observed data to this basis via ridge regression, then extrapolates the result to generate forecasts. Our regularization term encourages the time-indexed basis elements to be more unit standardized and less mutually correlated, intended to enable more robust ridge regression. The regularized variant matches or outperforms DeepTime on all LTSF benchmarks. Moreover, it is significantly more resilient to missing values in the lookback window at test time, enhances forecast accuracy when applied to higher-frequency data than it was trained on, and boosts performance when trained on smaller datasets. Overall, we conclude that our regularized approach sets a new state-of-the-art for deep time-index models.

## 1 Introduction

Time-series forecasting is a rich topic with applications spanning science (e.g., in weather or climate science, astronomy, biology (Ghaderpour et al., 2021; Rackauckas et al., 2020)), engineering (e.g., in control engineering, communications engineering (Krishna et al., 2023; Hua et al., 2017)), and business (e.g., in econometrics, mathematical finance, demand forecasting, capacity planning and management (Carbonneau et al., 2008; Laptev et al., 2017)). The rise of deep learning models and the increasing availability of large time-series datasets are stimulating a shift towards highly expressive time-series models calibrated using big data and big compute, rather than the classical approach of simpler models calibrated using strong assumptions and human expertise. However, leveraging large and expressive models based on deep learning for time-series forecasting has proven challenging. For example, Zeng et al. (2022) showed that many state-of-the-art

---

[†]Work was while the author was an intern at RBC Borealis

Transformer-based time-series models could be outperformed by "embarrassingly simple" linear models on common long time-series forecasting (LTSF) benchmarks.

Recently, the DeepTime model (Woo et al., 2023) demonstrated the promise of deep *time-index models*, as opposed to the more popular *historical-value models*. Historical-value models (e.g., ARIMA (Box et al., 2015) and popular Transformer-based models (Vaswani et al., 2017)) typically produce forecasts at pre-defined moment(s) in the future based on a pre-defined sequence of recent values, e.g., $\hat{y}_{t+\Delta t} = \phi(y_t, y_{t-1}, \ldots, y_{t-L})$, where $t$ is the current time, $\hat{y}_{t+\Delta t}$ is the model output for time $t + \Delta t$, $y_t$ denotes ground truth observations at time $t$, and $\phi$ is the function learned by the model. In contrast, time-index models are continuous functions of time, e.g., $\hat{y}_{t+\Delta t} = \phi(t + \Delta t)$. In this case, some parameters of the model are typically refit at inference so that $\hat{y}_{t+\Delta t} \approx y_{t+\Delta t}$ for $t + \Delta t$ values in the past, and predictions are made by extrapolating $\phi(t + \Delta t)$. Classical examples of such approaches include Prophet (Taylor & Letham, 2018) and Gaussian processes (Rasmussen, 2004). Time-index models have some appealing properties. For instance, the continuous dependence of $\phi$ on time is arguably a useful inductive bias for representing typical time-series data (Woo et al., 2023). Moreover, historical-value models often struggle with inconsistent input sequences or target horizons, while time-index models are well-suited for handling irregular sampling rates, missing values, and forecasts over a continuous horizon.

The core idea of DeepTime is to parameterize a time-indexed basis of features $f(\Delta t)$ by an implicit neural representation (INR) network. However, simply fitting a generic neural network to past values of $y$ at inference time is slow and leads to poor extrapolation into the future. DeepTime overcomes this issue by framing forecasting as a meta-learning problem: globally, learn a basis of time-indexed features $f(\Delta t)$ such that, locally, forecasting can be done by simple ridge regression. The INR training objective is the forecasting error, with the goal of finding a basis that can simultaneously (a) fit lookback values in a reliable manner and (b) extrapolate that fit to the forecast period accurately. This formulation achieves results competitive with other deep forecasting models on common LTSF benchmarks (Woo et al., 2023).

In this work, we improve upon DeepTime via a regularization term designed to better condition the ridge regression stage as shown in Figure 1. Essentially, we refine the meta-learning objective to be: globally, learn a basis of *normalized and uncorrelated* time-indexed features such that, locally, forecasting can be done *robustly* by simple ridge regression. As widely understood by practitioners, and provably true under certain assumptions (Krikheli & Leshem, 2021), linear regression performs better when the explanatory variables are more uniformly normalized and less mutually correlated. Our regularizer is simple, efficient, and encourages these properties in the learned INR basis. Our model code is available at `https://github.com/BorealisAI/DeepRRTime`. We call our robust, regularized DeepTime method DeepRRTime and summarize our contributions as follows:

1. We introduce a novel regularization technique for improving the forecasting accuracy of Deep-Time models. Our regularization technique is theoretically motivated, easy to implement, and computationally inexpensive.

2. We demonstrate empirically that the proposed regularizer facilitates the learning of more unit-normalized and less mutually correlated basis representations, improving performance on several benchmarks at little cost and setting a new state-of-the-art for time-index models.

3. We show that the proposed regularizer helps to improve forecast accuracy in more challenging settings: (a) forecasting with missing values, (b) training on smaller datasets, (c) forecasting at a higher-frequency at test-time compared to the training data.

## 2 Related work

### 2.1 Models for time-series forecasting

Classically, time-series forecasting has typically been conducted using simple models and strong assumptions about the underlying data generation process, relying on human expertise. Famous examples of classical historical-value models include exponential smoothing models like ETS (Holt, 2004; Hyndman, 2018) and the ARIMA family of models (Box et al., 2015). Classical time-series models in the class of time-index models (as defined by Woo et al. (2023)) include the Prophet algorithm (Taylor & Letham, 2018) and Gaussian processes (Rasmussen, 2003). DeepTime (Woo et al., 2023) and TimeFlow (Naour et al., 2024) are recent

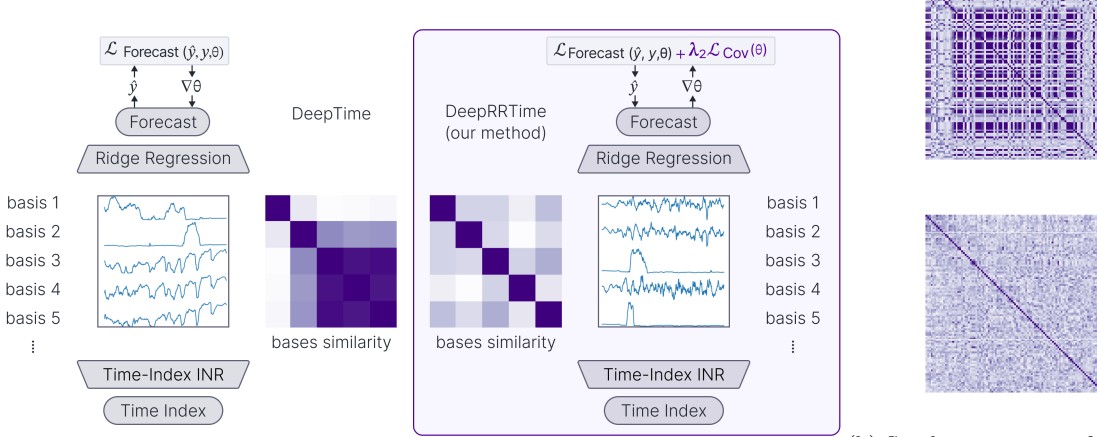

(a) Comparison of our approach with DeepTime.

(b) Similarity matrices for DeepTime (top) and DeepRRTime (bottom).

Figure 1: (a): Our proposed approach is similar to DeepTime, which takes time indices as input, maps them into a basis, and applies ridge regression to predict future time series. However, DeepRRTime includes an additional regularization term added to the forecast loss, promoting more standardized and less correlated time-indexed basis elements. This regularization leads to more distinct basis elements, resulting in better conditioning for ridge regression and more robust forecasting. (b): Cosine similarity matrices for 100 of the 256 basis elements (truncated for visibility), demonstrating that the basis learned by DeepRRTime is much closer to orthonormal. The top matrix shows the similarity between the bases learned by DeepTime, while the bottom matrix illustrates the similarity between the bases learned by DeepRRTime on the same dataset (ETTm2, forecast horizon 96).

models that utilize time-index models for time-series forecasting, and DeepTime in particular was the first demonstration of a deep time-index model with performance competitive to historical-value models.

The time-series forecasting research community has shown significant interest in historical-value models based on deep learning, with the hope that their increased expressivity will capture more complex relationships in data. Oreshkin et al. (2019) proposed N-BEATS, a deep learning model based on a trend and seasonality decomposition, while Challu et al. (2023) introduced N-HiTS, which employs hierarchical interpolation and multi-rate sampling. LogTrans (Li et al., 2019), FEDformer (Zhou et al., 2022), and ETSFormer (Woo et al., 2022) models attempt to leverage attention-based architectures (Vaswani et al., 2017) to model dependencies across time steps. Adaptive RNNs (Du et al., 2021), RevIN (Kim et al., 2021), and Non-stationary Transformers (Liu et al., 2022) introduce various methods aimed at handling distribution shifts.

Zeng et al. (2022) recently published simple linear models that outperform all the Transformer-based models listed above on all the standard LTSF benchmarks, and argued against applying attention layers to individual time steps. PatchTST (Nie et al., 2023) effectively addressed this concern, segmenting time-series data into patches to address the potential lack of semantic meaning in individual time steps. PatchTST achieved state-of-the-art performance on almost all the LTSF benchmark problems, matching or outperforming the linear baselines of (Zeng et al., 2022). However, one notable exception is the Exchange dataset, which was omitted from the PatchTST experiments. The authors highlighted it as singularly challenging among all the benchmarks, and noted that even a simple martingale model is a formidable baseline to overcome for this dataset.

## 2.2 Meta-learning

Meta-learning, or the "learning to learn" paradigm, entails training across multiple tasks to learn to generalize to new, unseen tasks. The widely recognized MAML (Finn et al., 2017) algorithm consists of an outer-step optimization spanning multiple tasks and an inner-step optimization on each task. The goal of the inner step is to perform well on a query set by fitting a neural network to a corresponding support set. The outer step aims to learn a neural network initialization such that, for all tasks, a small amount of fine-tuning on the support

set is sufficient to perform well on the query set. Reptile (Nichol et al., 2018) and ANIL (Raghu et al., 2020) demonstrated that MAML's expensive inner-loop fine-tuning can often be replaced by cheaper approximate or partial fine-tuning without hindering performance. Bertinetto et al. (2019a) eschewed the expensive iterative inner step entirely by employing a closed-form ridge regression solver for the classification head in a meta-learning model, significantly enhancing their efficiency. In DeepTime's meta-learning framework, each forecast is considered a distinct task. The support sets correspond to the recently observed data in the lookback window, and the query sets are the future values in the horizon window. In this framework, learning the INR basis is the outer loop, while ridge regression is the inner loop. As in Bertinetto et al. (2019a), DeepTime employs a closed-form solution to the inner-loop optimization, improving efficiency.

While we are not aware of previous work applying covariance regularization to meta-learning frameworks, as proposed in this paper, we note that covariance regularization has been considered for supervised and self-supervised tasks in computer vision (Cogswell et al., 2016; Zbontar et al., 2021; Bardes et al., 2022).

### 2.3 Implicit Neural Representation

Implicit neural representations (INR) are deep neural networks trained to approximate various types of signal including views of a scene (Mildenhall et al., 2021), images (Chen et al., 2021; Dupont et al., 2021), and 3D shapes (Chen & Zhang, 2019). For example, an INR trained to reconstruct an image might map a two-dimensional vector representing coordinates of an image pixel to a three-dimensional vector representing RGB colors at this pixel. Many INR architectures are relatively simple, consisting of simple variations of multilayer perceptrons (MLPs). SIRENs, proposed by (Sitzmann et al., 2020b), are MLPs with sinusoidal activation functions, chosen to facilitate the representation of high-frequency features. DeepTime uses a variant of the INR design proposed in (Tancik et al., 2020). Similar to SIRENs, the architecture of (Tancik et al., 2020) is also designed to capture high-frequency features, but uses a single Fourier embedding layer at the beginning of the network rather than periodic activations after each layer. Besides forecasting, INRs have been explored for other time-series applications: Jeong & Shin (2022) utilize INR representation errors as a metric for anomaly detection while Fons et al. (2022) and Szatkowski et al. (2023) explore applications to time-series generative-modeling and missing value imputation respectively. Meta-learning on INRs, as done in DeepTime, has been interpreted as learning a prior over the space of signals (Sitzmann et al., 2020a; Tancik et al., 2021; Dupont et al., 2022) and shown to have analogies to dictionary learning (Yüce et al., 2022).

## 3 Background

In this section, we review DeepTime and then present our method DeepRRTime. We first describe inference, which is the same for both approaches, and then contrast their training procedures.

### 3.1 Notation

Consider an INR network $f_\theta : \tau \mapsto z_\tau$ where $\tau \in [0, 1]$ is called the time-index and $z_\tau \in \mathbb{R}^D$ represents the values of the $D$-dimensional basis at time $\tau$. The time index $\tau$ is divided between the lookback and forecast regions as follows: given a lookback length $L$ and a forecast horizon $H$, the range $0 \leq \tau \leq \frac{L-1}{L+H-1}$ indexes the lookback window while $\frac{L}{L+H-1} \leq \tau \leq 1$ indexes the forecast period. Specifically, $\tau = \frac{L}{L+H-1}$ is the moment at which a forecast is to be made, in keeping with the convention of Woo et al. (2023).

We denote observed values of the target variable $y$ in the lookback window by $\mathbf{Y}_\ell = \{y_{\tau_1}, y_{\tau_2}, \ldots y_{\tau_\ell}\}$, with $\tau_i < \tau_j$ when $i < j$, $\tau_1 \geq 0$, and $\tau_\ell \leq \frac{L-1}{L+H-1}$. For regularly sampled data, we specialize this to $\tau_1 = 0$, $\tau_2 = \frac{1}{L+H-1}$, and so on to $\tau_\ell = \frac{L-1}{L+H-1}$. We will refer to the model's forecast by $\hat{y}(\tau)$ for $\frac{L}{L+H-1} \leq \tau \leq 1$. We will generally consider a discrete set of forecasts $\widehat{\mathbf{Y}}_h = \{\hat{y}_{\tau_{\ell+1}}, \hat{y}_{\tau_{\ell+2}}, \ldots \hat{y}_{\tau_{\ell+h}}\}$. For regularly sampled data, we specialize this to $\tau_{\ell+1} = \frac{L}{L+H-1}$, $\tau_{\ell+2} = \frac{L+1}{L+H-1}$, and so on to $\tau_{\ell+h} = 1$.

### 3.2 Inference of DeepTime and DeepRRTime

The inference step of our method DeepRRTime is essentially identical to that of the original DeepTime method, which we repeat here. Inference begins with the evaluation of the INR basis at the time index values for which lookback data are available: $\mathbf{Z}_\ell = \{z_{\tau_1}, z_{\tau_2}, \ldots z_{\tau_\ell}\} = \{f_\theta(\tau_1), f_\theta(\tau_2), \ldots, f_\theta(\tau_\ell)\}$. We then "explain the past" by solving the following system of equations for $W$ and $b$: $\mathbf{Y}_\ell = \mathbf{Z}_\ell W + b$. Specifically, we solve for the ridge-regression optimal parameters:

$$W^*, b^* = \underset{W,b}{\arg\min} ||\mathbf{Y}_\ell - \mathbf{Z}_\ell W - b||_2^2 + \lambda_1 \left( ||W||_2^2 + ||b||_2^2 \right), \tag{1}$$

where $\lambda_1$ is an L2 penalty coefficient.

The parameters $(W^*, b^*)$ which solve the ridge regression problem over the lookback window can be obtained with the closed-form solution $(W^*, b^*) = (\tilde{\mathbf{Z}}_\ell^T \tilde{\mathbf{Z}}_\ell + \lambda_1 I)^{-1} \tilde{\mathbf{Z}}_\ell^T \mathbf{Y}_\ell$ where $\tilde{\mathbf{Z}}_\ell = [\mathbf{Z}_\ell; \mathbf{1}]$ is obtained by concatenating a vector of ones. They are then used to extrapolate $\hat{y}(\tau) = z_\tau W^* + b^*$ into the forecast region and generate the set of predictions:

$$\widehat{\mathbf{Y}}_h = \mathbf{Z}_h W^* + b^*, \tag{2}$$

where $\mathbf{Z}_h = \{z_{\tau_{\ell+1}}, z_{\tau_{\ell+2}}, \ldots z_{\tau_{\ell+h}}\}$.

### 3.3 Training of DeepTime

We begin by reviewing the training procedure for DeepTime, and later explain how our method DeepRRTime differs. The core idea is that the meta-learner, $f_\theta$, is exposed during training to the way in which the base learner, ridge regression, solves for the optimal parameters $(W^*, b^*)$. The meta-learning routine can therefore learn an INR basis that allows the base learner to forecast the future reliably.

DeepTime optimizes the INR $f_\theta$ to learn a basis $\{z_\tau\}$ that, through the above inference procedure, minimizes the forecasting error. Specifically, given $\mathbf{Y}_\ell$, $\mathbf{Z}_\ell$ and $\mathbf{Z}_h$, we first compute $(W^*, b^*)$ and then use Equation 2 to compute $\widehat{\mathbf{Y}}_h$. We then optimize the INR parameters, $\theta$, to minimize the forecast mean squared error (MSE):

$$\mathcal{L}_{\text{Forecast}}(y, \hat{y}, \theta) = \arg\min_\theta ||\mathbf{Y}_h - \widehat{\mathbf{Y}}_h(\theta, W^*(\theta), b^*(\theta))||_2^2, \tag{3}$$

where $\mathbf{Y}_h$ represents the ground-truth observations at the same time indices as $\widehat{\mathbf{Y}}_h$. Note that the inner optimization step itself depends on $\theta$, as implied by the notation $(W^*(\theta), b^*(\theta))$. The closed form solution to ridge regression enables DeepTime to propagate efficient, high-quality gradients through the entire inner optimization step. The meta-learning step is key to DeepTime's adaptation for time-series forecasting: among all possible representations, the INR seeks to learn one that allows the ridge regression solver to explain the past values, $\mathbf{Y}_\ell$, in such a way as to reliably forecast the future values, $\mathbf{Y}_h$.

## 4 DeepRRTime: regularized basis for improved forecasting

In this section, we introduce a regularization technique to improve the forecasting accuracy of the DeepTime model by refining the basis functions learned by the INR. Informally, we consider the question of how to learn temporal patterns in the basis elements that will improve the forecasting accuracy. Since "good" temporal patterns are highly dependent on the specific forecasting task, it is difficult to define them in a domain-agnostic way. Therefore, we approach this problem through the lens of the ridge regression employed by DeepTime, asking instead how to learn basis elements that are *better suited for ridge regression*. In the following, we identify key properties of such a basis and propose a regularization objective to encourage these properties during the training process.

We first note the sharp contrast between our method and traditional regularization techniques to improve the conditioning of linear regression on correlated variates, e.g., (Adnan & Hura Ahmad, 2006; Paul, 2006; Herawati et al., 2018; Chan et al., 2022). As a rule, such methods assume that the correlated variates are specified externally to the problem, and then seek to process them one way or another to minimize the impact of the collinearity on the regression process. In contrast, the INR representation in DeepTime is entirely learned; there are no externally specified collinear variates to disentangle. Our method exploits the unique freedom in this setting to construct these variates specifically such that they lead to a well-conditioned regression problem.

Linear regression problems have been the subject of numerous theoretical analyses owing to their simplicity and efficiency. In the following, we use the theoretical results from Krikheli & Leshem (2021) to develop an understanding of how the basis influences the prediction errors. Theorem III.1 in (Krikheli & Leshem, 2021) suggests that the smallest and the largest eigenvalues ($\lambda_{\min}$ and $\lambda_{\max}$) of the sample covariance matrix influence the sample-efficiency for a required error threshold. Therefore, for a given number of train samples, a basis whose covariance matrix has larger $\lambda_{\min}$ and smaller $\lambda_{\max}$ would result in lower prediction error. Motivated by this observation, we propose to improve the ridge regression optimization by controlling the eigenvalues of the covariance matrix through a regularization term.

In standard applications, linear regression is applied directly to observed data with a fixed data-determined covariance matrix. In contrast, the basis $z_\tau$ in our case is learned, enabling direct control of the covariance matrix properties. While it is possible in principle to directly minimize the largest eigenvalue and maximize the smallest eigenvalue of the sample covariance matrix, the key parameters in the theorems of Krikheli & Leshem (2021), we have found it more practical to make two variations as discussed below.

### 4.1 Centered covariance matrix regularizer

Motivated by the theory of Krikheli & Leshem (2021) on robust regression, we aim to develop a regularizer that encourages the INR to learn a basis with a better-conditioned covariance matrix. Instead of regularizing the *uncentered* covariance matrix, which arises directly in the theory, we instead regularize the *centered* covariance matrix,

$$G_\theta = \frac{1}{L+H}\mathbf{Z}\mathbf{Z}^\top - \mu\mu^\top, \tag{4}$$

where $G_\theta(ij)$ indicates the covariance between the $i$-th and $j$-th basis elements, $\mathbf{Z} = [\mathbf{Z}_\ell; \mathbf{Z}_h]$ is the concatenation of the lookback and forecast bases and $\mu = \frac{1}{L+H}\sum_\tau z_\tau \in \mathbb{R}^D$ is the mean along the temporal axis for each of the $D$ basis dimensions. Compared to regularizing the uncentered covariance matrix, this approach leaves the absolute means of the basis elements unconstrained; preliminary experiments suggested this flexibility was advantageous to the learning process.

From the Weyl inequalities, we show below that controlling the eigenvalues of the centered covariance matrix will also control *most* of the eigenvalues of the uncentered covariance matrix. The exception is the largest eigenvalue, which can be poorly controlled if the basis elements are large.

**Theorem 1.** *(Weyl, 1912) For Hermitian matrices $A, B \in \mathbb{C}^n$, the eigenvalues of $A + B$ is related to the eigenvalues of the individual matrices as follows:*

$$\lambda_{i+j-1}(A+B) \leq \lambda_i(A) + \lambda_j(B), \quad i+j \leq n+1 \tag{5}$$
$$\lambda_i(A) + \lambda_j(B) \leq \lambda_{i+j-n}(A+B), \quad i+j \geq n+1 \tag{6}$$

*where $i,j = 1,\ldots,n$ and $\lambda_n \leq \lambda_{n-1} \leq \cdots \leq \lambda_1$.*

Specifically, if $A = G_\theta$, $B = \mu\mu^\top$ and $A + B = \frac{1}{L+H}\mathbf{Z}\mathbf{Z}^\top$, we can find upper- and lower-bounds for the eigenvalues of the uncentered covariance matrix[*]. Considering $i = n$ and $j = n$ in Equation (6), we conclude that $\lambda_n(G_\theta) \leq \lambda_n\left(\frac{1}{L+H}\mathbf{Z}\mathbf{Z}^\top\right)$. That is, the smallest eigenvalue of the uncentered covariance matrix (which must not be too small, according to the theory of Krikheli & Leshem (2021)) is lower-bounded by the smallest eigenvalue of the centered covariance $G_\theta$. Thus, it is sufficient to control the smallest eigenvalue via $G_\theta$. Considering $j = 1$ and $i = 1$ in Equation (5), we see that $\lambda_1\left(\frac{1}{L+H}\mathbf{Z}\mathbf{Z}^\top\right) \leq \lambda_1(G_\theta) + \mu^\top\mu$. That is, the largest eigenvalue of the uncentered covariance matrix can exceed that of $G_\theta$ by $\mu^\top\mu$ for basis means $\mu$.

In practice, we have found that the INR generally does not appear to learn basis elements with pathologically large means. The INR initialization and training protocols do not seem conducive to representing such functions *a priori*, and we would expect the forecast error term to be sufficient to prevent them from emerging during training. Conversely, as noted above, our early experiments empirically showed that the extra flexibility of unconstrained basis means appears to be very advantageous to the training process. Thus, we find that regularizing $G_\theta$ instead of the uncentered covariance matrix is the more practical approach.

### 4.2 Indirect regularization of eigenvalues

In our preliminary experiments, directly regularizing the largest and smallest eigenvalues of $G_\theta$ led to instabilities in the optimization process. Instead, we found it more tractable and effective to regularize $G_\theta$ towards the identity matrix, thereby regularizing all of its eigenvalues towards 1, using a term of the form:

$$\mathcal{L}_{\text{Cov}}(\theta) = \frac{1}{D^2}\left[\sum_{1 \leq i \neq j \leq D} G_\theta(ij)^2 + \sum_{1 \leq i \leq D}(G_\theta(ii) - 1)^2\right] = \frac{1}{D^2}\|G_\theta - I\|_F^2, \tag{7}$$

where $\|\cdot\|_F$ stands for Frobenius norm. Intuitively, the first sum in $\mathcal{L}_{\text{Cov}}(\theta)$ penalizes non-zero covariances between elements in the centered basis, while the second sum encourages the variances of each element in the

---

[*]Note that since $(\mu\mu^\top)\mu = \mu(\mu^\top\mu) = (\mu^\top\mu)\mu$, and since $\mu\mu^\top$ is rank one, the only non-zero eigenvalue of $\mu\mu^\top$ is $\mu^\top\mu \geq 0$.

centered basis to be close to 1. Therefore, when $\mathcal{L}_{\text{Cov}}(\theta)$ is small the centered basis is closer to orthonormal, and it is orthonormal if and only if $G_\theta$ equals the identity matrix. The fact that all the eigenvalues of $G_\theta$ lie within an interval near 1, with the size of the interval upper-bounded in proportion to $\mathcal{L}_{\text{Cov}}(\theta)$, directly follows from Theorem 2 and we refer the reader to Appendix A for further details.

**Theorem 2.** *(Gershgorin, 1931); or, e.g., (Johnson & Horn, 1985) for an English presentation.* *Let $A$ be a complex $n \times n$ matrix with entries $a_{ij}$. For $i \in \{1, \ldots, n\}$ let $R_i$ be the sum of the absolute values of the non-diagonal entries in the $i$-th row:*

$$R_i = \sum_{j \neq i} |a_{ij}|. \tag{8}$$

*Let $D(a_{ii}, R_i) \subset \mathbf{C}$ be a closed disc centered at $a_{ii}$ with radius $R_i$. Such a disc is called a Gershgorin disc.*

*Then every eigenvalue of $A$ lies within at least one of the Gershgorin discs $D(a_{ii}, R_i)$.*

Although the theory of Krikheli & Leshem (2021) does not directly require all eigenvalues to be near 1 for regression to be robust, this is certainly a sufficient condition for their robustness results to apply. Specifically, this theory's regression error bounds diverge as the smallest eigenvalue of the uncentered covariance matrix approaches zero, which is an outcome that $\mathcal{L}_{\text{Cov}}(\theta)$ directly discourages.

### 4.3 Summary of DeepRRTime

In summary, our proposed method, DeepRRTime, differs from the original framework of DeepTime by altering the primary objective for the outer loop of training to the following:

$$\mathcal{L}_{\text{Forecast}}(y, \hat{y}, \theta) + \lambda_2 \mathcal{L}_{\text{Cov}}(\theta), \tag{9}$$

where $\lambda_2$ is the covariance regularizer coefficient.

Constraining $\mathcal{L}_{\text{Cov}}(\theta)$ provably improves the conditioning of the INR basis. The smallest eigenvalue of the *uncentered* covariance matrix is discouraged from being too small, with an upper bound of its distance below 1 controlled in proportion to $\mathcal{L}_{\text{Cov}}(\theta)$. The largest eigenvalue of the *centered* covariance matrix is also bounded near 1, with the largest eigenvalue of the *uncentered* covariance matrix growing only insofar as the basis means are large (which does not appear to occur in practice). As we show in Figure 6 in the Appendix, minimizing Equation (9) does reduce $\mathcal{L}_{\text{Cov}}$ of the DeepRRTime model substantially compared to DeepTime. In fact, $\mathcal{L}_{\text{Cov}}$ grows with training epochs for DeepTime, suggesting that the basis is becoming increasingly ill-conditioned over time. In the following section, we demonstrate empirical evidence to support these views: DeepTime indeed exhibits the instabilities that one would expect from a correlated basis, whereas DeepRRTime exhibits the robustness we anticipated from a basis better suited for regression.

## 5 Results

In this section, we evaluate the effectiveness of our regularization method, DeepRRTime, on real-world datasets and compare it with relevant baselines. In addition to the regular timeseries forecasting problem, we also test its robustness in three more challenging settings to illuminate advantages of the proposed regularization technique: forecasting with missing values in the lookback window, training with reduced dataset size, and forecasting on a finer time grid at test time than the one used during training.

### 5.1 Experimental setup

For evaluation, we use 6 real-world benchmarks for LTSF which include Electricity Transformer Temperature (ETTm2), Electricity Consumption Load (ECL), Exchange, Traffic, Weather, and Influenza-like Illness (ILI), detailed in Appendix B. For each dataset, we evaluate our model on tasks with four distinct forecast horizons, as is standard practice in the related literature. We also use the standard train, validation, and test sets splits: 60/20/20 for ETTm2 and 70/10/20 for the remaining 5 datasets (Woo et al., 2023). We preprocess each dataset by standardization based on train set statistics. To compare our method with other approaches, we employ two metrics: mean squared error (MSE) and mean absolute error (MAE). For each experiment, we report error statistics across 10 random network initializations: only average errors are reported in the main text, with standard deviations included in the Appendix. Hyperparameter selection, when applied, is performed for the lookback length multiplier $\mu$ which defines the length of the lookback window $L$ relative to the prediction horizon $H$ as $L = \mu H$. We search through the values $\mu \in \{1, 3, 5, 7, 9\}$, and select the best value based on the validation loss. Throughout this work, we use identical values for all common

hyperparameters of DeepTime and DeepRRTime besides the lookback multiplier $\mu$. We recompute the results of DeepTime using the original open source code implementation, and report errors over 10 random network initializations. We emphasize that our implementation is largely based on the original code of DeepTime and the proposed regularization requires minimal code changes. Please refer to Appendix C for a complete list of hyper parameters and further implementation details.

## 5.2 Multivariate forecasting

In Table 1, we consider the regular multivariate time-series forecasting problem and compare our proposed approach, DeepRRTime, with other time-index models for multivariate time-series forecasting including TimeFlow (Naour et al., 2024) and Gaussian Processes (GP) (Rasmussen, 2004). We emphasize the comparison of DeepRRTime with DeepTime in particular, to understand the relative benefits of our proposed regularization. We also compare DeepRRTime to historical-value models including recent Transformer-based architectures such as PatchTST (Nie et al., 2023) as well as NLinear, a linear timeseries forecasting model, shown to be competitive with many of the far more computationally expensive Transformer models (Zeng et al., 2022). Lastly, we include the performance of a simple martingale model that outputs the last observed value as a forecast and was shown to be competitive on some LTSF benchmarks. Additionally, Tables 5a and 5b in the Appendix extends the comparison of DeepRRTime to a broader set of historical-value models including NS Transformer (Liu et al., 2022), N-HiTS (Challu et al., 2023), ETSFormer (Woo et al., 2022), FEDformer (Zhou et al., 2022), CrossFormer (Wang et al., 2022), TimesNet (Wu et al., 2023) and iTransformer (Liu et al., 2023). All experiments in this section are conducted with the regularization coefficient fixed to $\lambda_2 = 1$, as jointly optimizing $\lambda_2$ with the lookback multiplier $\mu$ for the full set of benchmarks was prohibitively expensive.

| | | Time-index models | | | | | | | | Historical-value models | | | | | |
| | | DeepRRTime | | DeepTime | | GP | | TimeFlow | | NLinear | | Martingale | | PatchTST | |
| Metrics | | MSE | MAE | MSE | MAE | MSE | MAE | MSE | MAE | MSE | MAE | MSE | MAE | MSE | MAE |
|---|---|---|---|---|---|---|---|---|---|---|---|---|---|---|---|
| ETTm2 | 96 | **0.166** | **0.258** | **0.166** | **0.258** | 0.442 | 0.422 | 0.269 | 0.322 | 0.167 | 0.255 | 0.266 | 0.328 | 0.166 | 0.256 |
| | 192 | 0.224 | 0.300 | **0.223** | **0.299** | 0.605 | 0.505 | 0.394 | 0.399 | 0.221 | 0.293 | 0.340 | 0.371 | 0.223 | 0.296 |
| | 336 | **0.276** | **0.338** | 0.278 | 0.339 | 0.731 | 0.569 | 0.523 | 0.471 | 0.274 | 0.327 | 0.412 | 0.410 | 0.274 | 0.329 |
| | 720 | **0.368** | **0.397** | 0.383 | 0.411 | 0.959 | 0.669 | 0.663 | 0.557 | 0.368 | 0.384 | 0.521 | 0.465 | 0.362 | 0.385 |
| ECL | 96 | **0.137** | **0.238** | **0.137** | **0.238** | 0.503 | 0.538 | 0.141 | 0.240 | 0.141 | 0.237 | 1.588 | 0.946 | 0.129 | 0.222 |
| | 192 | **0.152** | **0.251** | **0.152** | 0.252 | 0.505 | 0.543 | 0.155 | **0.251** | 0.154 | 0.248 | 1.595 | 0.950 | 0.147 | 0.240 |
| | 336 | **0.165** | **0.267** | 0.166 | 0.268 | 0.612 | 0.614 | 0.170 | 0.268 | 0.171 | 0.265 | 1.617 | 0.961 | 0.163 | 0.259 |
| | 720 | **0.202** | 0.303 | **0.202** | 0.302 | 0.652 | 0.635 | 0.203 | **0.300** | 0.210 | 0.297 | 1.647 | 0.975 | 0.197 | 0.290 |
| Exchange | 96 | **0.078** | **0.197** | 0.079 | 0.199 | 0.136 | 0.267 | 0.307 | 0.395 | 0.089 | 0.208 | 0.081 | 0.196 | 0.088* | 0.207* |
| | 192 | 0.153 | **0.284** | **0.152** | 0.285 | 0.229 | 0.348 | 1.450 | 0.658 | 0.180 | 0.300 | 0.167 | 0.289 | 0.191* | 0.312* |
| | 336 | **0.257** | **0.375** | 0.324 | 0.424 | 0.372 | 0.447 | 3.691 | 1.063 | 0.331 | 0.415 | 0.305 | 0.396 | 0.358* | 0.436* |
| | 720 | **0.541** | **0.540** | 0.675 | 0.592 | 1.135 | 0.810 | 8.184 | 1.626 | 1.033 | 0.780 | 0.823 | 0.681 | 0.932* | 0.728* |
| Traffic | 96 | **0.390** | **0.274** | **0.390** | **0.274** | 1.112 | 0.665 | 2.623 | 0.287 | 0.410 | 0.279 | 2.723 | 1.079 | 0.360 | 0.249 |
| | 192 | **0.402** | **0.278** | **0.402** | **0.278** | 1.133 | 0.671 | 5.621 | 0.305 | 0.423 | 0.284 | 2.756 | 1.087 | 0.379 | 0.256 |
| | 336 | **0.416** | **0.285** | **0.416** | 0.289 | 1.274 | 0.723 | 23.648 | 0.331 | 0.435 | 0.290 | 2.791 | 1.095 | 0.392 | 0.264 |
| | 720 | **0.450** | **0.307** | **0.450** | 0.308 | 1.280 | 0.719 | 15.013 | 0.357 | 0.464 | 0.307 | 2.811 | 1.097 | 0.432 | 0.286 |
| Weather | 96 | **0.166** | **0.222** | 0.167 | 0.223 | 0.395 | 0.356 | 0.186 | 0.242 | 0.182 | 0.232 | 0.259 | 0.254 | 0.149 | 0.198 |
| | 192 | **0.207** | **0.260** | **0.207** | **0.260** | 0.450 | 0.398 | 0.252 | 0.299 | 0.225 | 0.269 | 0.309 | 0.292 | 0.194 | 0.241 |
| | 336 | **0.251** | **0.298** | 0.252 | 0.300 | 0.508 | 0.440 | 0.318 | 0.343 | 0.271 | 0.301 | 0.377 | 0.338 | 0.245 | 0.282 |
| | 720 | **0.312** | **0.348** | 0.313 | 0.350 | 0.498 | 0.450 | 0.393 | 0.394 | 0.338 | 0.348 | 0.465 | 0.394 | 0.314 | 0.334 |
| ILI | 24 | **2.317** | 1.044 | 2.558 | 1.115 | 2.331 | **1.036** | 3.199 | 1.228 | 1.683 | 0.858 | 6.587 | 1.701 | 1.319 | 0.754 |
| | 36 | 2.253 | 1.022 | 2.264 | 1.042 | **2.167** | **1.002** | 3.166 | 1.212 | 1.703 | 0.859 | 7.130 | 1.884 | 1.579 | 0.870 |
| | 48 | **2.292** | 1.033 | 2.302 | **1.027** | 2.961 | 1.180 | 3.128 | 1.180 | 1.719 | 0.884 | 6.575 | 1.798 | 1.553 | 0.815 |
| | 60 | 2.301 | 1.035 | **2.292** | **1.030** | 3.108 | 1.214 | 3.563 | 1.277 | 1.819 | 0.917 | 5.893 | 1.677 | 1.470 | 0.788 |

Table 1: Comparison of DeepRRTime with time-index models on multivariate benchmarks for long sequence time-series forecasting. For reference, we also provide the corresponding results for representative historical-value models. Best results for time-index models are **bolded**, and best results overall are colored in blue. *The results of PatchTST on the Exchange dataset were not reported by the authors and are therefore computed by us while the remaining results are obtained from the respective papers.

We summarize our findings from these experiments as follows:

(a) **DeepRRTime matches or exceeds the performance of other time-index models overall.** DeepRRTime achieves the best performance out of the time-index models in 38 out of 48 settings in

Table 1. Moreover, considering the standard deviations and the fourth digit of precision reported in Table 6 in the Appendix, we find that the cases where DeepTime outperforms DeepRRTime are generally not statistically significant: only the difference in MSE on ECL/720 comes close, with a difference slightly exceeding two standard errors. Similarly, the underperformance relative to GP on ILI/24 is not statistically significant, although the underperformance relative to TimeFlow in 1 of 2 metrics on ECL/720 and relative to GP on ILI/36 are significant. Altogether, accounting for measurement uncertainty, DeepRRTime matches or exceeds the other time-index models in 45 out of 48 results, and matches or exceeds DeepTime in all settings. Conversely, 17 of the cases where DeepRRTime outperforms DeepTime are statistically significant as determined by a one-sided Welch's t-test with a significance level of 0.05. Most notably, the covariance regularization introduces significant improvements on the Exchange dataset, especially on the longest forecast horizons (336, 720) where we achieve between 10% and 20% improvement in MAE and MSE over DeepTime. Altogether, these results suggest that the DeepRRTime regularizer is a clear improvement to DeepTime's already state-of-the-art performance among time-index models.

(b) **DeepRRTime achieves state-of-the-art performance on the challenging Exchange dataset and performs comparably with historical-value models on other datasets.** Despite the improvements of DeepRRTime over DeepTime, PatchTST outperforms both almost uniformly on 5 out of 6 of the LTSF benchmarks. We find this suggestive of a strong inductive bias in the PatchTST model that is very beneficial to modelling many types of time-series data. However, we see that both DeepTime and DeepRRTime outperform PatchTST by a large margin on the Exchange dataset, suggesting this same inductive bias can be counterproductive for some data types. The Exchange dataset is considered to be a very challenging LTSF benchmark due to its low signal-to-noise ratio and its low degree of stationarity (see Table 8 in the Appendix reporting the ADF test statistic (Liu et al., 2022), a measure of non-stationarity). Indeed, the authors of PatchTST (Nie et al., 2023) acknowledged omitting Exchange from their experiments for this reason. Besides PatchTST, Table 5a in the Appendix shows that DeepRRTime outperforms the other Transformer-based historical-value models uniformly on ECL, Exchange, and Traffic, and on nearly all the ETTm2 metrics, although results are more mixed on Weather and ILI datasets.

The results of this section demonstrate that DeepRRTime is a state-of-the-art time-index model, and that time-index models can be competitive with state-of-the-art historical-value models in at least some cases. Although PatchTST still exhibits a strong advantage in predictive performance on many LTSF datasets, it is also worth noting that time-index models have other practical advantages. For one, time-index models are generally less computationally expensive, especially at inference time (in particular, DeepRRTime is much more efficient than PatchTST as shown in Table 11 in Appendix). The next several sections explore use cases highlighting other putative advantages of time-index models over historical-value models, and demonstrate that the regularization added to DeepRRTime is key to capitalizing on these advantages.

| Evaluation | | No missing values | | | | 50% missing lookback values | | | |
|---|---|---|---|---|---|---|---|---|---|
| | | Average improvement | | Highest improvement | | Average improvement | | Highest improvement | |
| Train portion | Data | MSE | MAE | MSE | MAE | MSE | MAE | MSE | MAE |
| | ETTm2 | 1.05% | 0.84% | 3.92% | 3.41% | 4.15% | 3.21% | 8.66% | 5.77% |
| | ECL | 0.15% | 0.11% | 0.60% | 0.40% | 0.50% | 0.56% | 0.94% | 0.93% |
| Full | Exchange | 10.28% | 5.42% | 20.68% | 11.56% | 24.46% | 11.58% | 53.66% | 23.52% |
| | Traffic | 0.00% | 0.43% | 0.00% | 1.38% | 0.44% | 0.88% | 0.71% | 1.60% |
| | Weather | 0.33% | 0.42% | 0.60% | 0.67% | -0.11% | 0.12% | 0.50% | 0.72% |
| | ILI | 2.49% | 1.80% | 9.42% | 6.37% | 1.85% | 0.99% | 6.84% | 3.97% |
| | ETTm2 | 22.65% | 16.39% | 41.94% | 28.80% | 23.75% | 16.92% | 42.33% | 29.20% |
| 10% of data | ECL | 14.64% | 13.07% | 20.40% | 16.96% | 10.13% | 8.52% | 16.59% | 12.12% |
| | Traffic | 0.43% | 1.21% | 0.65% | 1.94% | 0.52% | 0.96% | 0.68% | 0.99% |
| | Weather | 40.53% | 31.11% | 60.36% | 46.70% | 46.92% | 37.62% | 60.95% | 47.89% |

Table 2: Summary of improvements introduced with the proposed regularizer (i.e., DeepTime vs DeepRRTime) for all combinations of training (e.g., full-data vs 10% of training data) and evaluation (e.g., 50% missing lookback values vs no missing values). We observe that the regularization technique introduces higher improvements for more challenging settings of train and/or evaluation. For each dataset, we average results over 4 forecast horizons and report average and highest relative improvements in terms of MSE and MAE.

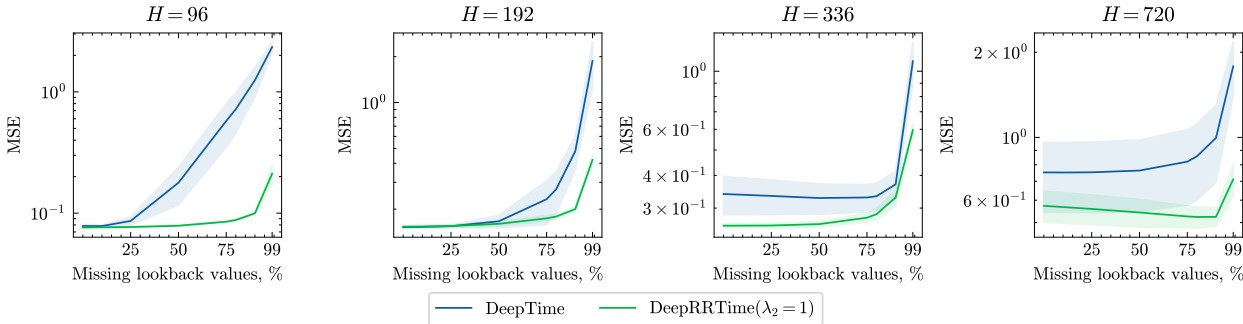

Figure 2: Plots of mean squared error (MSE) of DeepRRTime and DeepTime as a function of missing lookback values percentage for different forecast horizons on Exchange dataset. The shadow areas report standard deviations over 10 network initializations. While the performance of DeepTime deteriorates significantly for higher missing rates, our model exhibits a remarkable level of robustness to the missing values, even when 90% of the values are missing.

| Methods | | DeepTime | | DeepRRTime | | DeepRRTime | |
|---|---|---|---|---|---|---|---|
| | | 50% missing lookback values | | | | No missing values | |
| Metrics | | MSE | MAE | MSE | MAE | MSE | MAE |
| ETTm2 | 96 | 0.200 | 0.301 | **0.183** | **0.284** | 0.165 | 0.258 |
| | 192 | 0.233 | 0.315 | **0.230** | **0.312** | 0.224 | 0.300 |
| | 336 | 0.282 | 0.347 | **0.278** | **0.343** | 0.276 | 0.337 |
| | 720 | 0.383 | 0.414 | **0.363** | **0.394** | 0.368 | 0.397 |
| ECL | 96 | 0.155 | 0.263 | **0.154** | **0.262** | 0.136 | 0.237 |
| | 192 | 0.166 | 0.274 | **0.165** | **0.272** | 0.151 | 0.250 |
| | 336 | 0.180 | 0.289 | **0.179** | **0.287** | 0.165 | 0.266 |
| | 720 | 0.215 | 0.319 | **0.214** | **0.319** | 0.201 | 0.302 |
| Exchange | 96 | 0.175 | 0.268 | **0.081** | **0.205** | 0.077 | 0.197 |
| | 192 | 0.166 | 0.303 | **0.158** | **0.292** | 0.152 | 0.284 |
| | 336 | 0.311 | 0.416 | **0.259** | **0.379** | 0.256 | 0.375 |
| | 720 | 0.665 | 0.593 | **0.516** | **0.532** | 0.541 | 0.539 |

| Methods | | DeepTime | | DeepRRTime | | DeepRRTime | |
|---|---|---|---|---|---|---|---|
| | | 50% missing lookback values | | | | No missing values | |
| Metrics | | MSE | MAE | MSE | MAE | MSE | MAE |
| Traffic | 96 | **0.417** | 0.295 | **0.417** | **0.293** | 0.389 | 0.273 |
| | 192 | 0.424 | 0.297 | **0.422** | **0.295** | 0.401 | 0.278 |
| | 336 | 0.438 | 0.305 | **0.435** | **0.300** | 0.416 | 0.285 |
| | 720 | 0.475 | 0.325 | **0.473** | **0.322** | 0.450 | 0.307 |
| Weather | 96 | **0.176** | **0.242** | 0.177 | 0.243 | 0.166 | 0.222 |
| | 192 | **0.216** | **0.275** | 0.218 | 0.277 | 0.207 | 0.260 |
| | 336 | 0.262 | 0.314 | **0.260** | **0.312** | 0.251 | 0.298 |
| | 720 | 0.318 | 0.359 | **0.316** | **0.357** | 0.312 | 0.348 |
| ILI | 24 | 2.571 | 1.120 | **2.395** | **1.076** | 2.317 | 1.043 |
| | 36 | 2.309 | 1.046 | **2.291** | **1.033** | 2.252 | 1.022 |
| | 48 | 2.352 | **1.032** | **2.344** | 1.039 | 2.291 | 1.033 |
| | 60 | **2.328** | **1.033** | 2.341 | 1.039 | 2.301 | 1.034 |

Table 3: Our covariance regularization introduces further improvements upon DeepTime when forecasting with 50% of lookback values missing. For reference, we also include the results of DeepRRTime when the whole lookback window is available. See Table 9 for the standard-deviations.

### 5.3 Missing values in the lookback window

We now explore a more challenging setting to showcase additional benefits of the covariance regularization: forecasting with missing observations in the lookback window. In contrast to historical-value models, which typically assume a fixed lookback window, time-index models can, in principle, handle missing values in lookback windows without any extra architectural modifications even when trained with regularly-sampled training data (i.e., no missing values). While other model families, such as Neural ODEs (Rubanova et al., 2019) and Graph Networks (Yalavarthi et al., 2024), have been designed to handle irregular time-series, we do not compare to them in this work and focus instead on time-index models. We will show in this section, however, that although DeepTime is technically capable of making forecasts in spite of missing values, it does not perform very well in this setting. In contrast, we find DeepRRTime is extremely robust to missing values in the lookback window, consistent with the expected advantages of learning a better-conditioned basis.

For this experiment, we do not change the training of DeepTime and DeepRRTime, but randomly mask out 50% of the samples in the lookback window at test time when computing the regression coefficients $W^*$ and $b^*$ (Equation 1). Table 3 presents the performance of both models in this setting, compared to the performance of DeepRRTime without missing lookback values. We notice that DeepRRTime is more resilient to this test than DeepTime on all datasets except Weather; on the shortest two horizons of Weather, DeepRRTime underperforms DeepTime by a statistically significant but nonetheless small amount. Importantly, the missing-

**(a) No missing lookback values**

| | | 10% of data | | | | Full data | |
| | | DeepTime | | DeepRRTime | | DeepRRTime | |
| Methods | Metrics | MSE | MAE | MSE | MAE | MSE | MAE |
|---|---|---|---|---|---|---|---|
| ETTm2 | 96 | 0.210 | 0.306 | **0.181** | **0.276** | 0.165 | 0.258 |
| | 192 | 0.285 | 0.357 | **0.241** | **0.313** | 0.224 | 0.300 |
| | 336 | 0.374 | 0.412 | **0.301** | **0.351** | 0.276 | 0.337 |
| | 720 | 0.653 | 0.566 | **0.379** | **0.403** | 0.368 | 0.397 |
| ECL | 96 | 0.200 | 0.313 | **0.159** | **0.260** | 0.136 | 0.237 |
| | 192 | 0.209 | 0.323 | **0.175** | **0.276** | 0.151 | 0.250 |
| | 336 | 0.231 | 0.343 | **0.193** | **0.292** | 0.165 | 0.266 |
| | 720 | 0.264 | 0.365 | **0.249** | **0.342** | 0.201 | 0.302 |
| Traffic | 96 | 0.414 | 0.288 | **0.411** | **0.283** | 0.389 | 0.273 |
| | 192 | 0.428 | 0.302 | **0.425** | **0.299** | 0.401 | 0.278 |
| | 336 | 0.446 | 0.298 | **0.446** | **0.295** | 0.416 | 0.285 |
| Weather | 96 | 0.273 | 0.346 | **0.188** | **0.252** | 0.166 | 0.222 |
| | 192 | 0.394 | 0.435 | **0.256** | **0.317** | 0.207 | 0.260 |
| | 336 | 0.535 | 0.509 | **0.344** | **0.388** | 0.251 | 0.298 |
| | 720 | 0.900 | 0.713 | **0.357** | **0.380** | 0.312 | 0.348 |

**(b) 50% missing lookback values**

| | | 10% of data | | | | Full data | |
| | | DeepTime | | DeepRRTime | | DeepRRTime | |
| Methods | Metrics | MSE | MAE | MSE | MAE | MSE | MAE |
|---|---|---|---|---|---|---|---|
| ETTm2 | 96 | 0.214 | 0.309 | **0.180** | **0.277** | 0.183 | 0.284 |
| | 192 | 0.287 | 0.359 | **0.239** | **0.312** | 0.230 | 0.312 |
| | 336 | 0.374 | 0.412 | **0.299** | **0.350** | 0.278 | 0.343 |
| | 720 | 0.652 | 0.565 | **0.376** | **0.400** | 0.363 | 0.394 |
| ECL | 96 | 0.217 | 0.330 | **0.181** | **0.290** | 0.154 | 0.262 |
| | 192 | 0.217 | 0.331 | **0.192** | **0.299** | 0.165 | 0.272 |
| | 336 | 0.240 | 0.351 | **0.220** | **0.325** | 0.179 | 0.287 |
| | 720 | 0.270 | 0.370 | **0.259** | **0.352** | 0.214 | 0.319 |
| Traffic | 96 | 0.438 | 0.303 | **0.435** | **0.300** | 0.417 | 0.293 |
| | 192 | 0.448 | 0.315 | **0.445** | **0.312** | 0.422 | 0.295 |
| | 336 | 0.473 | 0.316 | **0.472** | **0.313** | 0.435 | 0.300 |
| Weather | 96 | 0.313 | 0.389 | **0.185** | **0.249** | 0.177 | 0.243 |
| | 192 | 0.425 | 0.464 | **0.237** | **0.293** | 0.218 | 0.277 |
| | 336 | 0.543 | 0.518 | **0.317** | **0.364** | 0.260 | 0.312 |
| | 720 | 0.886 | 0.710 | **0.346** | **0.370** | 0.316 | 0.357 |

Table 4: The proposed regularizer further improves forecast upon DeepTime when models are trained using 10% of the most recent data. For reference, we also provide the results when using the full training dataset.

value experiment helps to reveal differences between DeepTime and DeepRRTime that were not evident in the regular forecasting setup — in Table 2, aggregating the improvements of DeepRRTime over DeepTime in different scenarios, we generally observe that the difference between DeepTime and DeepRRTime widens in the missing-value setting as compared to the default setting: examples where average MSE improves are ECL (0.15%→0.50%), Traffic (0.00%→0.44%), ETTm2 (1.05%→4.15%) and Exchange (10.28%→24.46%). Based on the Welch's one-sided t-test, we find that DeepRRTime achieves statistically significant improvements in 30 out of 40 metrics on the missing-value test. Interestingly, the state-of-the-art performance of DeepRRTime on the longer forecasting horizons of Exchange barely degrades at all, compared to its performance without masked inputs, illustrating the robustness conferred by our regularization.

We further extend our analysis and plot MSE of DeepRRTime and DeepTime with missing rates of 25%, 50%, 75%, 90%, and 99% on the Exchange dataset in Figure 2. The plots show that while the MSE of DeepTime grows up to an order of magnitude with increasing masking rate, DeepRRTime exhibits little degradation even when 90% of the lookback values are missing. Also see Figure 8 in Appendix for results extended to other datasets.

In summary, the regularization we have added to DeepRRTime enables robust forecasting in the presence of missing lookback values. This is a putative advantage of time-index models over historical-value models, but our experiments reveal that the original DeepTime model does not actually perform stably in this setting. We also reiterate that most historical-value models, including PatchTST, cannot natively operate with lookback window samples that differ from the exact sequence on which they were trained ( for e.g., see Table 12 for an evaluation of Patch-TST with missing-values using linear-interpolation/zero-substitution to handle missing values): the forward evaluation of these models is simply undefined for sequences of a different length. We expect that the robustness of DeepRRTime to missing samples in the lookback window should be particularly useful in certain real-world scenarios where only irregularly sampled values are available at test time.

## 5.4 Smaller training dataset size

A common method to evaluate a new regularization technique is to test its ability to improve generalization when training with smaller datasets (Srivastava et al., 2014). To create smaller training datasets, we select the last 10% of the training set, resulting in 10× fewer observations. As this leads to reduced updates per epoch as compared to the default training scenario, we increase the number of epochs, early-stopping patience and warmup epochs by 10×. We normalize the data using mean and standard deviation statistics estimated using the reduced training dataset. To obtain error metrics comparable to the other settings, we report MSE and MAE obtained by renormalizing model outputs using statistics estimated using the full training dataset.

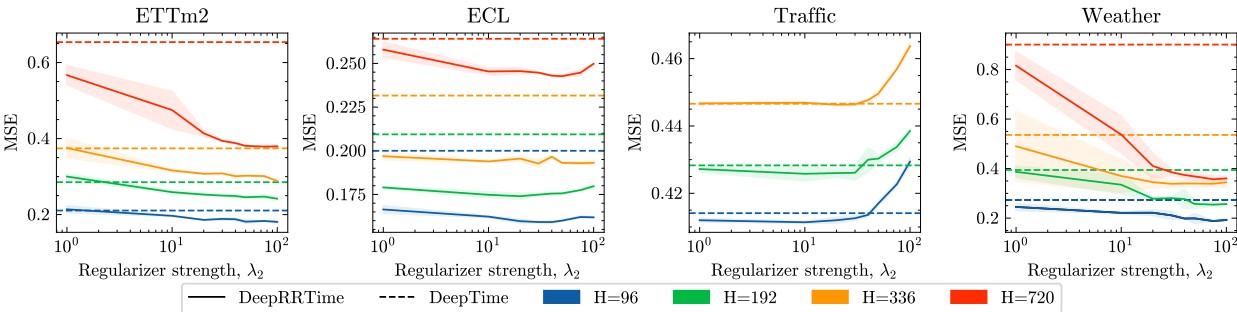

Figure 3: Plots showing influence of the regularizer strength $\lambda_2$ on MSE when models are trained on 10% of the most recent data. The results are shown for different datasets and forecast horizons (color-coded according to the legend). The baseline performance of DeepTime is shown with dashed lines, while performance of DeepRRTime is shown with solid lines. The shadow areas report standard deviations over 10 network initializations. We observe that the covariance regularization introduces significant improvements with increasing regularization strength. Note that $\lambda_2$ is shown on a logarithmic-scale.

We exclude the Exchange and ILI datasets as well as Traffic/720 from this experiment, since reducing these datasets $10\times$ leaves too few training samples to form a single training batch. In this section, we evaluate DeepRRTime by selecting $\lambda_2$ from $\{1, 10, 25, 50, 75, 100\}$ based on the validation loss.

Table 4 presents a comparison between DeepRRTime and DeepTime when trained on 10% of data for both the settings without missing lookback values and with 50% of lookback values missing. As observed in Table 4a, DeepTime experiences a significant performance drop when trained on reduced datasets. In contrast, our regularization considerably narrows this performance gap, with DeepRRTime outperforming DeepTime by approximately 20%, and the average MSE of DeepRRTime increasing by 11.8% with reduced dataset sizes compared to a 30% increase in the case of DeepTime. A similar trend is observed in Table 4b when forecasting with missing lookback values. Figure 3 shows the performance of DeepRRTime on every dataset for different values of $\lambda_2$ varying from $\lambda_2 = 0$ (i.e., no regularization corresponding to DeepTime) to $\lambda_2 = 100$. These results suggest that, in most cases, tuning $\lambda_2$ for a specific problem improves performance, and that the results in Section 5.2 might be further improved if sufficient resources were allocated to training with different values of $\lambda_2$. Based on these experiments, we suggest that our regularization objective is particularly likely to add value over the unregularized DeepTime when training on relatively small training datasets.

## 5.5 Test-time interpolation

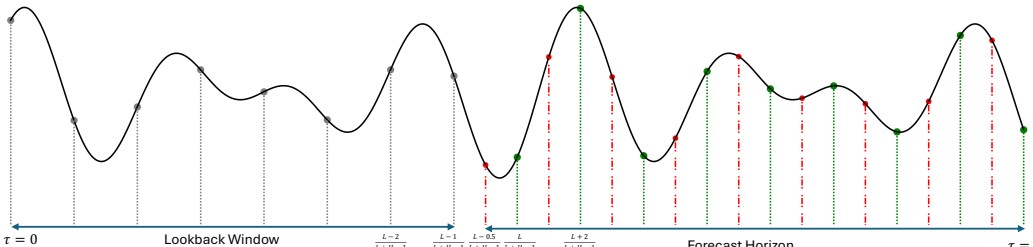

Figure 4: An illustration of the test-time interpolation setting where the time-index model is used to generate forecasts at $2\times$ the frequency it was trained at. At train time, the grey points on the time grid are used to estimate the linear-regression parameters while the green points on the time grid are used to compute the train loss. At test time, we exploit the flexibility of time-index models to interpolate between the training time grid (i.e., green points) to forecast at a higher frequency (i.e. both green and red points). Note that the red points on the time grid denote time indexes that are only seen at test time.

In this section, we consider the problem of forecasting at a greater frequency at test-time (e.g., every 30 minutes) using a model trained to forecast at a smaller frequency (e.g., every hour). Recall that the DeepTime model first solves the least-squares optimization problem defined in Equation (1) to compute the optimal parameters $W^*, b^*$ based on lookback observations; next, the optimal parameters are applied to the basis $z_\tau = f_\theta(\tau)$ to

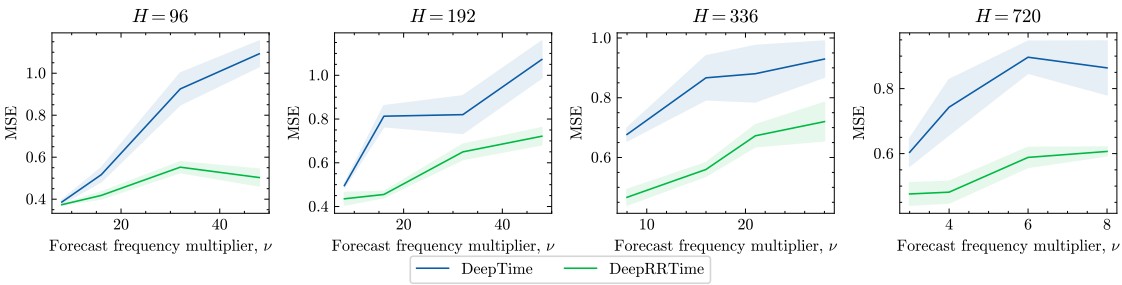

Figure 5: Comparison of our regularized method with the baseline on the test-time interpolation setting on the ETTm2 dataset. The shadow areas report standard deviations over 10 network initializations. We observe that DeepRRTime achieves significant improvements over DeepTime when forecasting at a frequency that is $\nu$ times higher at test time than the frequency observed during training.

generate the forecast for the following values of $\tau$: $\left\{\frac{L}{L+H-1}, \frac{L+1}{L+H-1}, \cdots 1\right\}$ where $L$ is the lookback window and $H$ is the forecast horizon. If the model is trained over hourly observations, then $\frac{1}{L+H-1}$ corresponds to a temporal difference of one hour. In order to use this model to generate a forecast at a frequency of 30 minutes, we could use the following sequence of time indexes instead: $\left\{\frac{L-0.5}{L+H-1}, \frac{L}{L+H-1}, \frac{L+0.5}{L+H-1}, \frac{L+1}{L+H-1}, \cdots 1\right\}$, see Figure 4 for an illustration. More generally, to forecast at an integer frequency $\nu$ higher at test time compared to train time, we apply the parameters $W^*, b^*$ over the basis representation obtained for the following time indexes: $\left\{\frac{\nu L-1}{\nu(L+H-1)}, \frac{\nu L}{\nu(L+H-1)}, \frac{\nu L+1}{\nu(L+H-1)}, \cdots 1\right\}$. Note that this setting interpolates between time indexes seen by the network at train time and evaluates the network in terms of its ability to generalize to novel time indexes seen only at test time.

To evaluate the model's ability to forecast at a higher frequency, we subsample the training data at different frequencies, using the ETTm2 dataset with observations spaced 15 minutes apart. For instance, when $\nu = 4$, the training data is subsampled to simulate observations at hourly intervals. At test time, we generate forecasts at the original higher frequency (e.g., every 15 minutes). We evaluate models on different combinations of $\nu$ and forecast horizons $H$, choosing $\mu \in \{1, 3, 5, 7, 9\}$ for both DeepTime and DeepRRTime and $\lambda_2 \in \{1, 10, 50\}$ for DeepRRTime based on the validation loss. The results presented in Figure 5 show that covariance regularization significantly improves performance across all $(\nu, H)$ pairs compared to DeepTime.

## 6 Conclusion

Our simple, inexpensive covariance regularizer leads to improvements on several LTSF benchmarks, including very significant improvements on the Exchange dataset. The enhanced robustness is particularly evident in the more challenging test settings: where lookback window values are missing at test time, forecasting is done at a higher frequency than training, or training is restricted to datasets with reduced sizes. These results suggest that regularization can greatly enhance deep time-index models. While we have only conducted preliminary hyperparameter tuning in this work, we expect that further tuning could potentially yield additional performance enhancement in practice.

## 7 Future Work

We believe that the results shown in our work open new opportunities for further research of deep time-index models. Moreover, we believe that the impact of our regularization method is not limited to the DeepTime model or time-series forecasting and consider the following directions for future work. Below, we list several directions for future work that we consider promising and important:

- **Efficient hyperparameter tuning.** Future work may consder exploring learning the optimal $\lambda_2$ and $\mu$ through gradient-based optimization or, more broadly speaking, employing techniques for automated hyperparameter tuning methods. Furthermore, developing techniques to enhance the computational efficiency of hyperparameter search presents a valuable future research direction in the field of time-index models as well as time-series forecasting, in general.

- **Direct control of covariance matrix eigenvalues.** Given the demonstrated efficacy of the simple covariance-regularization technique presented in our work, we believe that directly controlling the eigenvalues of the covariance matrix may be a fruitful direction for developing more powerful regularization techniques.
- **Non-linear generalization.** The benefits of the covariance regularization presented in our paper can motivate further exploration of time-index models employing non-linear base-learners (e.g., an MLP operating on the meta-learner's feature space) and regularization techniques that go beyond linear correlations.
- **Extensions to other applications.** We believe that our proposed regularization is not specific to time-series applications and can benefit other applications that use linear networks atop meta-learner's representations (e.g., (Bertinetto et al., 2019b)) which should be considered as future work.
- **Extending the evaluation to more benchmarks.** Given that deep time-index models is an emerging research area, it is important to understand their competitive advantages over better studied historical-value models by expanding the evaluation to diverse datasets: such as datasets with high frequency seasonal patterns, highly non-stationary behavior, or data from specific application domains.

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

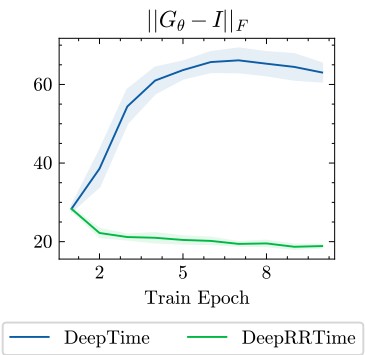

Figure 6: $\mathcal{L}_{\text{Cov}}[G_\theta] = \|G_\theta - I\|_F$ plotted across training epochs on the ETTm2 dataset with forecast horizon 96 and lookback multiplier $\mu = 1$. The shadow areas report standard deviations over 10 network initializations. While $\|G_\theta - I\|_F$ for DeepTime grows with epochs, our regularizer effectively decreases this value which results in a less mutually correlated basis. As predicted by theory discussed in Section 4, this leads to empirical improvements discussed throughout this section.

## A  Connection between the DeepRRTime regularizer and eigenvalues of $G_\theta$

By the Gershgorin circle theorem (Theorem 2), we see more precisely that when $\mathcal{L}_{\text{Cov}}(\theta)$ is small all the eigenvalues of $G_\theta$ will indeed be close to 1.

In the context of regularizing $G_\theta$ by $\mathcal{L}_{\text{Cov}}(\theta)$, note first that the radii of the Gershgorin discs of $G_\theta$ are constrained to be small when $\mathcal{L}_{\text{Cov}}(\theta)$ is small because

$$R_i^2[G_\theta] = \left( \sum_{1 \leq i \neq j \leq D} |G_\theta(ij)| \right)^2 \leq (D-1) \sum_{1 \leq i \neq j \leq D} G_\theta(ij)^2 \leq (D-1)D^2 \mathcal{L}_{\text{Cov}}(\theta). \tag{10}$$

Moreover, the center of each $i$-th disc is close to 1 according to

$$(G_\theta(ii) - 1)^2 \leq D^2 \mathcal{L}_{\text{Cov}}(\theta). \tag{11}$$

For sufficiently small $\mathcal{L}_{\text{Cov}}(\theta)$ (for any fixed choice of $D$) all eigenvalues therefore lie within Gershgorin discs having small radii and centers close to 1.

Since $G_\theta$ is Hermitian, its eigenvalues are real, and for small $\mathcal{L}_{\text{Cov}}(\theta)$ the eigenvalues must therefore lie in small intervals near 1. In particular, the smallest eigenvalue is bounded below by

$$\lambda_n(G_\theta) \geq 1 - \left( \sqrt{(D-1)D^2 \mathcal{L}_{\text{Cov}}(\theta)} + \sqrt{D^2 \mathcal{L}_{\text{Cov}}(\theta)} \right) = 1 - \left( \sqrt{D-1} + 1 \right) D \sqrt{\mathcal{L}_{\text{Cov}}(\theta)}, \tag{12}$$

so when $\sqrt{\mathcal{L}_{\text{Cov}}(\theta)} < 1/ \left( \sqrt{D-1} + 1 \right) D$ we have $\lambda_n(G_\theta) > 0$ and the basis is non-degenerate. Similarly, the largest eigenvalue of $G_\theta$ also cannot be very large when $\mathcal{L}_{\text{Cov}}(\theta)$ is small.

## B  Datasets

**ETTm2**[†] (Zhou et al., 2021) - Electricity Transformer Temperature dataset provides measurements from an electricity transformer such as load and oil temperature at a 15 minutes frequency.

**ECL**[‡] - The Electricity Consumption Load dataset comprises electricity usage data for 321 households, gathered between 2012 and 2014. Originally recorded every 15 minutes, the data is compiled into hourly aggregates.

**Exchange**[§] (Lai et al., 2018) - Dataset provides exchange rates of USD with currencies of eight countries (Australia, United Kingdom, Canada, Switzerland, China, Japan, New Zealand, and Singapore) from 1990 to 2016 at a daily frequency.

---

[†]https://github.com/zhouhaoyi/ETDataset
[‡]https://archive.ics.uci.edu/ml/datasets/ElectricityLoadDiagrams20112014
[§]https://github.com/laiguokun/multivariate-time-series-data

| Methods | | DeepRRTime | | PatchTST | | NS Transformer | | N-HiTS | | ETSformer | | FEDformer | | NLinear | | DLinear | | Martingale | |
|---|---|---|---|---|---|---|---|---|---|---|---|---|---|---|---|---|---|---|---|
| Metrics | | MSE | MAE | MSE | MAE | MSE | MAE | MSE | MAE | MSE | MAE | MSE | MAE | MSE | MAE | MSE | MAE | MSE | MAE |
| ETTm2 | 96 | **0.166** | 0.258 | **0.166** | 0.256 | 0.192 | 0.274 | 0.176 | **0.255** | 0.189 | 0.280 | 0.203 | 0.287 | 0.167 | **0.255** | 0.167 | 0.260 | 0.266 | 0.328 |
| | 192 | 0.224 | 0.300 | 0.223 | 0.296 | 0.280 | 0.339 | 0.245 | 0.305 | 0.253 | 0.319 | 0.269 | 0.328 | 0.221 | 0.293 | 0.224 | 0.303 | 0.340 | 0.371 |
| | 336 | 0.276 | 0.338 | 0.274 | 0.329 | 0.334 | 0.361 | 0.295 | 0.346 | 0.314 | 0.357 | 0.325 | 0.366 | 0.274 | 0.327 | 0.281 | 0.342 | 0.412 | 0.410 |
| | 720 | 0.368 | 0.397 | 0.362 | 0.385 | 0.417 | 0.413 | 0.401 | 0.426 | 0.414 | 0.413 | 0.421 | 0.415 | 0.368 | 0.384 | 0.397 | 0.421 | 0.521 | 0.465 |
| ECL | 96 | 0.137 | 0.238 | **0.129** | **0.222** | 0.169 | 0.273 | 0.147 | 0.249 | 0.187 | 0.304 | 0.183 | 0.297 | 0.141 | 0.237 | 0.140 | 0.237 | 1.588 | 0.946 |
| | 192 | 0.152 | 0.251 | **0.147** | **0.240** | 0.182 | 0.286 | 0.167 | 0.269 | 0.199 | 0.315 | 0.195 | 0.308 | 0.154 | 0.248 | 0.153 | 0.249 | 1.595 | 0.950 |
| | 336 | 0.165 | 0.267 | **0.163** | **0.259** | 0.200 | 0.304 | 0.186 | 0.290 | 0.212 | 0.329 | 0.212 | 0.313 | 0.171 | 0.265 | 0.169 | 0.267 | 1.617 | 0.961 |
| | 720 | 0.202 | 0.303 | **0.197** | **0.290** | 0.222 | 0.321 | 0.243 | 0.340 | 0.233 | 0.345 | 0.231 | 0.343 | 0.210 | 0.297 | 0.203 | 0.301 | 1.647 | 0.975 |
| Exchange | 96 | **0.078** | 0.197 | 0.088* | 0.207* | 0.111 | 0.237 | 0.092 | 0.211 | 0.085 | 0.204 | 0.139 | 0.276 | 0.089 | 0.208 | 0.081 | 0.203 | 0.081 | **0.196** |
| | 192 | **0.153** | **0.284** | 0.191* | 0.312* | 0.219 | 0.335 | 0.208 | 0.322 | 0.182 | 0.303 | 0.256 | 0.369 | 0.180 | 0.300 | 0.157 | 0.293 | 0.167 | 0.289 |
| | 336 | **0.257** | **0.375** | 0.358* | 0.436* | 0.421 | 0.476 | 0.371 | 0.443 | 0.348 | 0.428 | 0.426 | 0.464 | 0.331 | 0.415 | 0.305 | 0.414 | 0.305 | 0.396 |
| | 720 | **0.541** | **0.540** | 0.932* | 0.728* | 1.092 | 0.769 | 0.888 | 0.723 | 1.025 | 0.774 | 1.090 | 0.800 | 1.033 | 0.780 | 0.643 | 0.601 | 0.823 | 0.681 |
| Traffic | 96 | 0.390 | 0.274 | **0.360** | **0.249** | 0.612 | 0.338 | 0.402 | 0.282 | 0.607 | 0.392 | 0.562 | 0.349 | 0.410 | 0.279 | 0.410 | 0.282 | 2.723 | 1.079 |
| | 192 | 0.402 | 0.278 | **0.379** | **0.256** | 0.613 | 0.340 | 0.420 | 0.297 | 0.621 | 0.399 | 0.562 | 0.346 | 0.423 | 0.284 | 0.423 | 0.287 | 2.756 | 1.087 |
| | 336 | 0.416 | 0.285 | **0.392** | **0.264** | 0.618 | 0.328 | 0.448 | 0.313 | 0.622 | 0.396 | 0.570 | 0.323 | 0.435 | 0.290 | 0.436 | 0.296 | 2.791 | 1.095 |
| | 720 | 0.450 | 0.307 | **0.432** | **0.286** | 0.653 | 0.355 | 0.539 | 0.353 | 0.632 | 0.396 | 0.596 | 0.368 | 0.464 | 0.307 | 0.466 | 0.315 | 2.811 | 1.097 |
| Weather | 96 | 0.166 | 0.222 | **0.149** | 0.198 | 0.173 | 0.223 | 0.158 | **0.195** | 0.197 | 0.281 | 0.217 | 0.296 | 0.182 | 0.232 | 0.176 | 0.237 | 0.259 | 0.254 |
| | 192 | 0.207 | 0.260 | **0.194** | 0.241 | 0.245 | 0.285 | 0.211 | 0.247 | 0.237 | 0.312 | 0.276 | 0.336 | 0.225 | 0.269 | 0.220 | 0.282 | 0.309 | 0.292 |
| | 336 | 0.251 | 0.298 | **0.245** | 0.282 | 0.321 | 0.338 | 0.274 | 0.300 | 0.298 | 0.353 | 0.339 | 0.380 | 0.271 | 0.301 | 0.265 | 0.319 | 0.377 | 0.338 |
| | 720 | **0.312** | 0.348 | 0.314 | **0.334** | 0.414 | 0.410 | 0.351 | 0.353 | 0.352 | 0.388 | 0.403 | 0.428 | 0.338 | 0.348 | 0.323 | 0.362 | 0.465 | 0.394 |
| ILI | 24 | 2.317 | 1.044 | **1.319** | **0.754** | 2.294 | 0.945 | 1.862 | 0.869 | 2.527 | 1.020 | 2.203 | 0.963 | 1.683 | 0.858 | 2.215 | 1.081 | 6.587 | 1.701 |
| | 36 | 2.253 | 1.022 | **1.579** | 0.870 | 1.825 | **0.848** | 2.071 | 0.969 | 2.615 | 1.007 | 2.272 | 0.976 | 1.703 | 0.859 | 1.963 | 0.963 | 7.130 | 1.884 |
| | 48 | 2.292 | 1.033 | **1.553** | **0.815** | 2.010 | 0.900 | 2.346 | 1.042 | 2.359 | 0.972 | 2.209 | 0.981 | 1.719 | 0.884 | 2.130 | 1.024 | 6.575 | 1.798 |
| | 60 | 2.301 | 1.035 | **1.470** | **0.788** | 2.178 | 0.963 | 2.560 | 1.073 | 2.487 | 1.016 | 2.545 | 1.061 | 1.819 | 0.917 | 2.368 | 1.096 | 5.893 | 1.677 |

(a) Best results are highlighted in **bold**, and second best results are underlined. *The results of PatchTST on the Exchange dataset were not reported by the authors and are therefore computed by us while the remaining results are obtained from the respective papers.

| Methods | | DeepRRTime | | CrossFormer | | TimesNet | | iTransformer | | PatchTST | |
|---|---|---|---|---|---|---|---|---|---|---|---|
| Metrics | | MSE | MAE | MSE | MAE | MSE | MAE | MSE | MAE | MSE | MAE |
| ETTm2 | 96 | **0.166** | **0.258** | 0.287 | 0.366 | 0.187 | 0.267 | 0.18 | 0.264 | **0.166** | **0.256** |
| | 192 | **0.224** | **0.3** | 0.414 | 0.492 | 0.249 | 0.309 | 0.25 | 0.309 | **0.223** | **0.296** |
| | 336 | **0.276** | **0.338** | 0.597 | 0.542 | 0.321 | 0.351 | 0.311 | 0.348 | **0.274** | **0.329** |
| | 720 | **0.368** | **0.397** | 1.73 | 1.042 | 0.408 | 0.403 | 0.412 | 0.407 | **0.362** | **0.385** |
| ECL | 96 | **0.137** | **0.238** | 0.219 | 0.314 | 0.168 | 0.272 | 0.148 | 0.24 | **0.129** | **0.222** |
| | 192 | **0.152** | **0.251** | 0.231 | 0.322 | 0.184 | 0.289 | 0.162 | 0.253 | **0.147** | **0.240** |
| | 336 | **0.165** | **0.267** | 0.246 | 0.337 | 0.198 | 0.3 | 0.178 | 0.269 | **0.163** | **0.259** |
| | 720 | **0.202** | **0.303** | 0.28 | 0.363 | 0.22 | 0.32 | 0.225 | 0.317 | **0.197** | **0.290** |
| Exchange | 96 | **0.078** | **0.197** | 0.256 | 0.367 | 0.107 | 0.234 | 0.086 | 0.206 | 0.088 | 0.207 |
| | 192 | **0.153** | **0.284** | 0.47 | 0.509 | 0.226 | 0.344 | 0.177 | 0.299 | 0.191 | 0.312 |
| | 336 | **0.257** | **0.375** | 1.268 | 0.883 | 0.367 | 0.448 | 0.331 | 0.417 | 0.358 | 0.436 |
| | 720 | **0.541** | **0.54** | 1.767 | 1.068 | 0.964 | 0.746 | 0.847 | 0.691 | 0.932 | 0.728 |
| Traffic | 96 | **0.39** | 0.274 | 0.522 | 0.29 | 0.593 | 0.321 | 0.395 | **0.268** | **0.360** | **0.249** |
| | 192 | **0.402** | 0.278 | 0.53 | 0.293 | 0.617 | 0.336 | 0.417 | **0.276** | **0.379** | **0.256** |
| | 336 | **0.416** | 0.285 | 0.558 | 0.305 | 0.629 | 0.336 | 0.433 | **0.283** | **0.392** | **0.264** |
| | 720 | **0.45** | 0.307 | 0.589 | 0.328 | 0.64 | 0.35 | 0.467 | **0.302** | **0.432** | **0.286** |
| Weather | 96 | 0.166 | 0.222 | **0.158** | 0.23 | 0.172 | 0.22 | 0.174 | **0.214** | **0.149** | **0.198** |
| | 192 | 0.207 | 0.26 | **0.206** | 0.277 | 0.219 | 0.261 | 0.221 | **0.254** | **0.194** | **0.241** |
| | 336 | **0.251** | **0.298** | 0.272 | 0.335 | 0.28 | 0.306 | 0.278 | **0.296** | **0.245** | **0.282** |
| | 720 | **0.312** | **0.348** | 0.398 | 0.418 | 0.365 | 0.359 | 0.358 | **0.347** | 0.314 | **0.334** |

(b) Best results amongst DeepRRTime, CrossFormer, TimesNet and iTransformer are highlighted in **bold**, and second best results are underlined. The best results overall are highlighted in **blue**. The CrossFormer, TimesNet and iTransformer results in this table are copied directly from Liu et al. (2023).

Table 5: Comparison of DeepRRTime with historical-value models on multivariate forecasting benchmarks for long sequence time-series forecasting.

**Traffic**¶ - Dataset from the California Department of Transportation provides road occupancy rates from 862 sensors located on the freeways of the San Francisco Bay area at a hourly frequency.

**Weather**‖ - Dataset provides measurements of 21 meteorological indicators such as air temperature and humidity throughout 2020 at a 10 minute frequency from the Weather Station of the Max Planck Biogeochemistry Institute.

---

¶https://pems.dot.ca.gov/
‖https://www.bgc-jena.mpg.de/wetter/

| Methods | DeepTime | | DeepRRTime | |
|---|---|---|---|---|
| Metrics | MSE | MAE | MSE | MAE |
| ETTm2 96 | 0.1658±0.0009 | **0.2581**±0.0019 | **0.1656**±0.0008 | 0.2585±0.0021 |
| ETTm2 192 | **0.2227**±0.0019 | **0.2994**±0.0015 | 0.2241±0.0022 | 0.3004±0.0029 |
| ETTm2 336 | 0.2778±0.0049 | 0.3386±0.0039 | **0.2764**±0.0025 | **0.3378**±0.0030 |
| ETTm2 720 | 0.3830±0.0062 | 0.4114±0.0056 | **0.3681**±0.0037 | **0.3970**±0.0048 |
| ECL 96 | 0.1373±0.0002 | 0.2381±0.0004 | **0.1369**±0.0002 | **0.2375**±0.0003 |
| ECL 192 | 0.1523±0.0004 | 0.2517±0.0005 | **0.1517**±0.0003 | **0.2507**±0.0004 |
| ECL 336 | 0.1656±0.0006 | 0.2677±0.0009 | **0.1653**±0.0003 | **0.2669**±0.0004 |
| ECL 720 | **0.2015**±0.0002 | **0.3023**±0.0003 | 0.2018±0.0004 | 0.3025±0.0005 |
| Exchange 96 | 0.0786±0.0018 | 0.1993±0.0035 | **0.0775**±0.0005 | **0.1974**±0.0006 |
| Exchange 192 | **0.1519**±0.0015 | 0.2854±0.0019 | 0.1528±0.0032 | **0.2840**±0.0025 |
| Exchange 336 | 0.3245±0.0287 | 0.4241±0.0190 | **0.2568**±0.0087 | **0.3752**±0.0041 |
| Exchange 720 | 0.6751±0.2371 | 0.5918±0.1004 | **0.5411**±0.0664 | **0.5396**±0.0301 |
| Traffic 96 | 0.3902±0.0003 | 0.2744±0.0005 | **0.3899**±0.0005 | **0.2738**±0.0004 |
| Traffic 192 | **0.4016**±0.0004 | 0.2784±0.0005 | 0.4018±0.0004 | **0.2784**±0.0005 |
| Traffic 336 | **0.4160**±0.0021 | 0.2885±0.0019 | 0.4162±0.0008 | **0.2854**±0.0006 |
| Traffic 720 | 0.4505±0.0006 | 0.3078±0.0007 | **0.4502**±0.0008 | **0.3073**±0.0013 |
| Weather 96 | 0.1667±0.0012 | 0.2233±0.0015 | **0.1661**±0.0006 | **0.2223**±0.0010 |
| Weather 192 | **0.2070**±0.0010 | **0.2603**±0.0014 | 0.2073±0.0005 | 0.2603±0.0007 |
| Weather 336 | 0.2522±0.0010 | 0.3001±0.0009 | **0.2510**±0.0011 | **0.2983**±0.0016 |
| Weather 720 | 0.3131±0.0008 | 0.3501±0.0011 | **0.3121**±0.0007 | **0.3481**±0.0016 |
| ILI 24 | 2.5578±0.1427 | 1.1151±0.0357 | **2.3171**±0.1312 | **1.0437**±0.0486 |
| ILI 36 | 2.2642±0.1279 | 1.0417±0.0441 | **2.2527**±0.0597 | **1.0224**±0.0204 |
| ILI 48 | 2.3019±0.1443 | **1.0270**±0.0315 | **2.2919**±0.1645 | 1.0330±0.0419 |
| ILI 60 | **2.2921**±0.1186 | **1.0296**±0.0369 | 2.3014±0.1531 | 1.0349±0.0457 |

Table 6: Comparison of DeepRRTime with DeepTime on multivariate benchmarks for long sequence time-series forecasting. Best results are highlighted in **bold**. The table reports mean and standard deviation over 10 random network initializations.

| | Hyperparameter | Value |
|---|---|---|
| | Epochs | 50 |
| | Learning rate | 1e-3 |
| | $\lambda_1$ learning rate | 1.0 |
| | Warm up epochs | 5 |
| | Batch size | 256 |
| | Early stopping patience | 7 |
| Parameters inherited | Max gradient norm | 10.0 |
| from DeepTime | Layer size | 256 |
| | $\lambda_1$ initialization | 0.0 |
| | Scales | $[0.01, 0.1, 1, 5, 10, 20, 50, 100]$ |
| | Fourier features size | 4096 |
| | INR dropout rate | 0.1 |
| | Lookback length multiplier, $\mu$ | $\mu \in \{1, 3, 5, 7, 9\}$ |
| Our parameters | $\lambda_2$ | 1.0 |

Table 7: Hyperparameters used in our experiments. We emphasize that we do not change any of the parameters inherited from DeepTime.

| Dataset | Number of variables | Frequency | Number of samples | ADF test statistic |
|---------|---------------------|-----------|-------------------|--------------------|
| Exchange | 8 | 1 Day | 7,588 | -1.889 |
| ILI | 7 | 1 Week | 966 | -5.406 |
| ETTm2 | 7 | 15 Minutes | 69,680 | -6.225 |
| ECL | 321 | 1 Hour | 26,304 | -8.483 |
| Traffic | 862 | 1 Hour | 17,544 | -15.046 |
| Weather | 21 | 10 Minutes | 52,695 | -26.661 |

Table 8: Summary of LTSF datasets with their ADF test statistics (Liu et al., 2022) where smaller ADF means a more stationary dataset.

**ILI**** - Influenza-like Illness dataset provides ratio of patients seen with ILI and the total number of patients, collected by the Centers for Disease Control and Prevention of the United States between 2002 and 2021 at a weekly frequency.

Table 8 presents further characteristics of the datasets including number of variables and samples, frequency, and ADF test statistic.

## C   Implementation details

We use the code provided by the authors of DeepTime paper while only making minimal changes related to the proposed regularizer and the conditioning network. We do not change any of the hyperparameters inherited from DeepTime and provide their values in Table 7. Below we repeat the description given by the authors in their paper (Woo et al., 2023).

**DeepTime model hyperparameters.**   DeepTime is trained using Adam optimizer (Kingma & Ba, 2014) with a learning rate scheduler following a linear warm-up and a cosine annealing scheme. We use gradient clipping by norm. The ridge regressor regularization coefficient, $\lambda_1$, is trained at a higher learning rate compared to the rest of meta parameters. The model is trained with early stopping based on validation loss, with a fixed patience parameter defined as the number of epochs for which the loss can increase before the training is stopped. We learn the ridge regression regularization coefficient parameter and constrain it to positive values via a softplus function. We apply ReLU activation, Dropout (Srivastava et al., 2014), and LayerNorm (Ba et al., 2016) after each INR layer. The dimension of Fourier embedding layer of INR is defined independently of the size of other layers. Here we specify the total size of the Fourier embedding layer where the number of dimensions for each Fourier frequency scale is computed as the size of the layer divided by the number of scales. We refer the reader to (Woo et al., 2023) for complete details of the model.

**TimeFlow implementation and reproducibility.**   To compare our approach with another time-index model, TimeFlow (Naour et al., 2024), which only reports results on a subset of our benchmarks, we used the implementation opensourced by Naour et al. (2024). The original code loads data from pre-processed files and does not have an interface to read CSV files. We augmented the original implementation with DeepTime data loaders to read data directly from CSV files and did our best to redefine the concepts of batches and epochs to align with TimeFlow code's interpretation. However, we were unable to reproduce the results reported by the authors on the ECL and Traffic datasets. We shared our implementation with the authors and asked them to verify the correctness of our reproduction. The authors confirmed that our implementation was correct and identified the normalization procedure as the source of the performance difference. Upon further investigation, we found that the discrepancy arose from TimeFlow's use of both the train and test data to compute statistics (i.e., mean and variance) for input data pre-processing. Following standard practices, this paper reports the results of TimeFlow that were obtained while using train set statistics to normalize both train and test sets.

## D   Univariate forecasting

We evaluate our proposed approach in the unvariate setting where the model predicts the last variable of 2 multivariate datasets. We follow the literature and use the Exchange and ETTm2 datasets for this

---

**https://gis.cdc.gov/grasp/fluview/fluportaldashboard.html

| Methods | | DeepTime | | DeepRRTime | |
|---|---|---|---|---|---|
| Metrics | | MSE | MAE | MSE | MAE |
| ETTm2 | 96 | 0.2009±0.0129 | 0.3015±0.0151 | **0.1835**±0.0047 | **0.2841**±0.0063 |
| | 192 | 0.2336±0.0073 | 0.3153±0.0087 | **0.2303**±0.0048 | **0.3120**±0.0060 |
| | 336 | 0.2824±0.0063 | 0.3473±0.0061 | **0.2787**±0.0044 | **0.3434**±0.0054 |
| | 720 | 0.3834±0.0077 | 0.4145±0.0067 | **0.3634**±0.0028 | **0.3942**±0.0042 |
| ECL | 96 | 0.1551±0.0005 | 0.2638±0.0006 | **0.1545**±0.0005 | **0.2620**±0.0009 |
| | 192 | 0.1667±0.0006 | 0.2740±0.0008 | **0.1659**±0.0004 | **0.2727**±0.0006 |
| | 336 | 0.1808±0.0005 | 0.2897±0.0007 | **0.1791**±0.0004 | **0.2870**±0.0007 |
| | 720 | 0.2150±0.0003 | 0.3197±0.0004 | **0.2146**±0.0005 | **0.3192**±0.0006 |
| Exchange | 96 | 0.1750±0.0696 | 0.2683±0.0406 | **0.0811**±0.0007 | **0.2052**±0.0013 |
| | 192 | 0.1668±0.0106 | 0.3037±0.0087 | **0.1582**±0.0046 | **0.2927**±0.0052 |
| | 336 | 0.3114±0.0315 | 0.4162±0.0228 | **0.2594**±0.0091 | **0.3797**±0.0044 |
| | 720 | 0.6655±0.1931 | 0.5937±0.0848 | **0.5168**±0.0593 | **0.5320**±0.0291 |
| Traffic | 96 | 0.4174±0.0005 | 0.2951±0.0007 | **0.4172**±0.0007 | **0.2937**±0.0004 |
| | 192 | 0.4246±0.0005 | 0.2973±0.0004 | **0.4229**±0.0004 | **0.2955**±0.0003 |
| | 336 | 0.4385±0.0007 | 0.3054±0.0007 | **0.4354**±0.0004 | **0.3005**±0.0005 |
| | 720 | 0.4759±0.0011 | 0.3250±0.0011 | **0.4730**±0.0048 | **0.3223**±0.0044 |
| Weather | 96 | **0.1766**±0.0012 | **0.2423**±0.0016 | 0.1778±0.0013 | 0.2431±0.0023 |
| | 192 | **0.2168**±0.0017 | **0.2758**±0.0019 | 0.2184±0.0006 | 0.2772±0.0007 |
| | 336 | 0.2620±0.0016 | 0.3146±0.0012 | **0.2607**±0.0020 | **0.3127**±0.0028 |
| | 720 | 0.3183±0.0008 | 0.3596±0.0012 | **0.3168**±0.0014 | **0.3570**±0.0023 |
| ILI | 24 | 2.5713±0.1238 | 1.1205±0.0255 | **2.3953**±0.1274 | **1.0760**±0.0493 |
| | 36 | 2.3090±0.1166 | 1.0469±0.0430 | **2.2915**±0.0495 | **1.0338**±0.0173 |
| | 48 | 2.3526±0.1320 | **1.0326**±0.0294 | **2.3446**±0.1482 | 1.0395±0.0407 |
| | 60 | **2.3287**±0.0835 | **1.0335**±0.0283 | 2.3414±0.1340 | 1.0397±0.0411 |

Table 9: Comparison of DeepRRTime with DeepTime on multivariate benchmarks for long sequence time-series forecasting with 50% of lookback values missing. Best results are highlighted in **bold**. The table reports mean and standard deviation over 10 random network initializations.

| Methods | | DeepRRTime | | DeepTime | | N-HiTS | | ETSformer | | FEDformer | | N-BEATS | | DeepAR | | Prophet | | ARIMA | | GP | |
|---|---|---|---|---|---|---|---|---|---|---|---|---|---|---|---|---|---|---|---|---|---|---|---|
| Metrics | | MSE | MAE | MSE | MAE | MSE | MAE | MSE | MAE | MSE | MAE | MSE | MAE | MSE | MAE | MSE | MAE | MSE | MAE | MSE | MAE |
| ETTm2 | 96 | 0.071 | 0.192 | 0.072 | 0.194 | 0.066 | **0.185** | 0.080 | 0.212 | **0.063** | 0.189 | 0.082 | 0.219 | 0.099 | 0.237 | 0.287 | 0.456 | 0.211 | 0.362 | 0.125 | 0.273 |
| | 192 | 0.096 | 0.232 | 0.096 | 0.231 | **0.087** | **0.223** | 0.150 | 0.302 | 0.102 | 0.245 | 0.120 | 0.268 | 0.154 | 0.310 | 0.312 | 0.483 | 0.261 | 0.406 | 0.154 | 0.307 |
| | 336 | 0.121 | 0.265 | 0.120 | 0.264 | **0.106** | **0.251** | 0.175 | 0.334 | 0.130 | 0.279 | 0.226 | 0.370 | 0.277 | 0.428 | 0.331 | 0.474 | 0.317 | 0.448 | 0.189 | 0.338 |
| | 720 | 0.177 | 0.327 | 0.178 | 0.328 | **0.157** | **0.312** | 0.224 | 0.379 | 0.178 | 0.325 | 0.188 | 0.338 | 0.332 | 0.468 | 0.534 | 0.593 | 0.366 | 0.487 | 0.318 | 0.421 |
| Exchange | 96 | **0.086** | 0.226 | **0.086** | 0.225 | 0.093 | **0.223** | 0.099 | 0.230 | 0.131 | 0.284 | 0.156 | 0.299 | 0.417 | 0.515 | 0.828 | 0.762 | 0.112 | 0.245 | 0.165 | 0.311 |
| | 192 | **0.174** | 0.329 | **0.174** | 0.331 | 0.230 | **0.313** | 0.223 | 0.353 | 0.277 | 0.420 | 0.669 | 0.665 | 0.813 | 0.735 | 0.909 | 0.974 | 0.304 | 0.404 | 0.649 | 0.617 |
| | 336 | **0.302** | **0.445** | 0.308 | 0.452 | 0.370 | 0.486 | 0.421 | 0.497 | 0.426 | 0.511 | 0.611 | 0.605 | 1.331 | 0.962 | 1.304 | 0.988 | 0.736 | 0.598 | 0.596 | 0.592 |
| | 720 | 0.836 | 0.741 | 0.845 | 0.752 | **0.728** | **0.569** | 1.114 | 0.807 | 1.162 | 0.832 | 1.111 | 0.860 | 1.890 | 1.181 | 3.238 | 1.566 | 1.871 | 0.935 | 1.002 | 0.786 |

Table 10: Univariate forecasting benchmarks on long sequence time-series forecasting. Best results are highlighted in **bold**, and second best results are underlined.

experiment (Liu et al., 2022; Challu et al., 2023; Woo et al., 2023). We compare our approach with both time-index models including DeepTime, Prophet (Taylor & Letham, 2018), Gaussian Processes (GP) (Rasmussen, 2004), and with historical-value models including N-HiTS (Challu et al., 2023), ETSFormer (Woo et al., 2022), FEDformer (Zhou et al., 2022), DeepAR (Salinas et al., 2020), and ARIMA (Anderson & Kendall, 1976). We note that the baseline models Prophet, DeepAR, and ARIMA are strictly univariate. We set the lookback multiplier to $\mu = 1$ for this experiment and report the results and report results in Table 10. We observe that our approach achieves the best performance across the baselines on 4 out of 16 metrics and scores as the second one on 5 more metrics. At the same time, N-HiTS outperforms our method on all the metrics on the ETTm2 dataset which we believe to be explained by its inductive bias for smoother data.

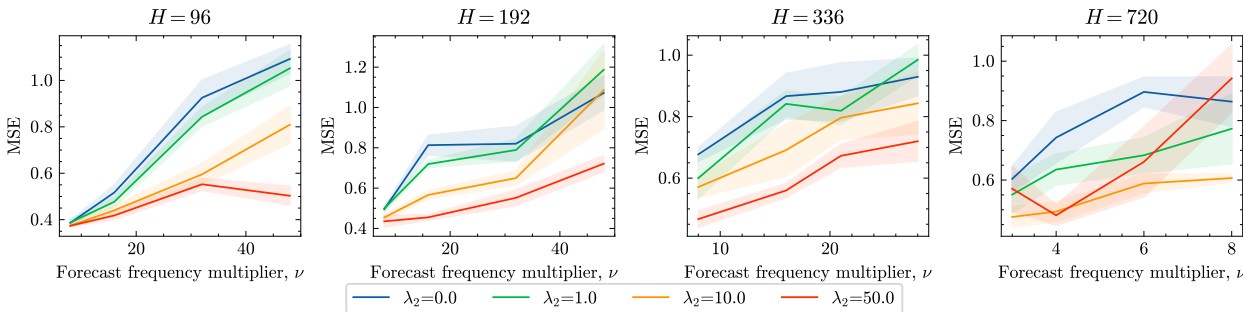

Figure 7: Performance of our regularized method with on the test-time interpolation setting on the ETTm2 dataset ($\lambda_2 = 0$ corresponds to DeepTime). The shadow areas report standard deviations over 10 network initializations. We observe that DeepRRTime achieves significant improvements when forecasting at an integer frequency $\nu$ higher at test time as compared the frequency observed at train time.

| Dataset/Horizon | Model | Training | | Inference | |
| --- | --- | --- | --- | --- | --- |
| | | Peak memory (GB) ↓ | Time per epoch (s) ↓ | Peak memory (GB) ↓ | Time per epoch (s) ↓ |
| Exchange/96 | DeepRRTime | **0.207** | **0.67** | **0.065** | **0.058** |
| | PatchTST | 0.449 | 3.68 | 0.153 | 1.45 |
| Exchange/720 | DeepRRTime | 2.98 | **1.88** | 0.774 | **0.257** |
| | PatchTST | **0.491** | 3.89 | **0.208** | 1.41 |
| ETTm2/96 | DeepRRTime | **1.05** | **6.39** | **0.448** | **1.43** |
| | PatchTST | 1.62 | 12.98 | 0.496 | 3.54 |
| ETTm2/720 | DeepRRTime | **1.48** | **8.13** | 1.32 | **2.8** |
| | PatchTST | 1.7 | 13.51 | **0.556** | 5.24 |
| Traffic/96 | DeepRRTime | **4.79** | **32.88** | **3.94** | **15.69** |
| | PatchTST | OOM | N/A | N/A | N/A |

Table 11: We measure the wall-clock time per epoch and peak memory usage in both train and inference modes when using PatchTST and DeepRRTime on a single NVIDIA Tesla V GPU (16GB). We observe that DeepRRTime is consistently faster than PatchTST in terms of both training and evaluation time. We also found DeepRRTime to be more memory efficient on average, however some exceptions exist such as Exchange/720. At the same time, we encountered a GPU out-of-memory error for PatchTST on all forecast horizons of Traffic while there was no such issue for DeepRRTime.

| | No missing lookback values | | | | 50% missing lookback values | | | | | |
| | PatchTST | | DeepRRTime | | PatchTST (replace with 0) | | PatchTST (linear interpolation) | | DeepRRTime | |
| $H$ | MSE | MAE | MSE | MAE | MSE | MAE | MSE | MAE | MSE | MAE |
|---|---|---|---|---|---|---|---|---|---|---|
| 96 | 0.1651±0.0010 | 0.2533±0.0011 | 0.1656±0.0008 | 0.2585±0.0021 | 0.9213±0.0580 | 0.6990±0.0211 | 0.6411±0.0478 | 0.5039±0.0157 | **0.1835±0.0047** | **0.2841±0.0063** |
| 192 | 0.2220±0.0009 | 0.2933±0.0007 | 0.2241±0.0022 | 0.3004±0.0029 | 0.9943±0.0253 | 0.7245±0.0096 | 0.5564±0.0388 | 0.4830±0.0141 | **0.2303±0.0048** | **0.3120±0.0060** |
| 336 | 0.2762±0.0009 | 0.3289±0.0007 | 0.2764±0.0025 | 0.3378±0.0030 | 1.0337±0.0247 | 0.7372±0.0077 | 0.5265±0.0417 | 0.4761±0.0160 | **0.2787±0.0044** | **0.3434±0.0054** |
| 720 | 0.3654±0.0013 | 0.3837±0.0006 | 0.3681±0.0037 | 0.3970±0.0048 | 1.0436±0.0446 | 0.7378±0.0160 | 0.5748±0.0696 | 0.5026±0.0268 | **0.3634±0.0028** | **0.3942±0.0042** |

Table 12: Comparison of DeepRRTime with PatchTST on ETTm2 with 50% of lookback values missing. We explore two techniques to enable PatchTST forecasting with missing values: (a) replacing missing values with 0, (b) linear interpolation. While we observe a significant drop of performance for PatchTST with both techniques, DeepRRTime remains robust on this cahllenging setting which highlights its advantage over historical-valued models. Best results with the missing values are highlighted in **bold**. The table reports means and standard deviations over 10 network initializations.

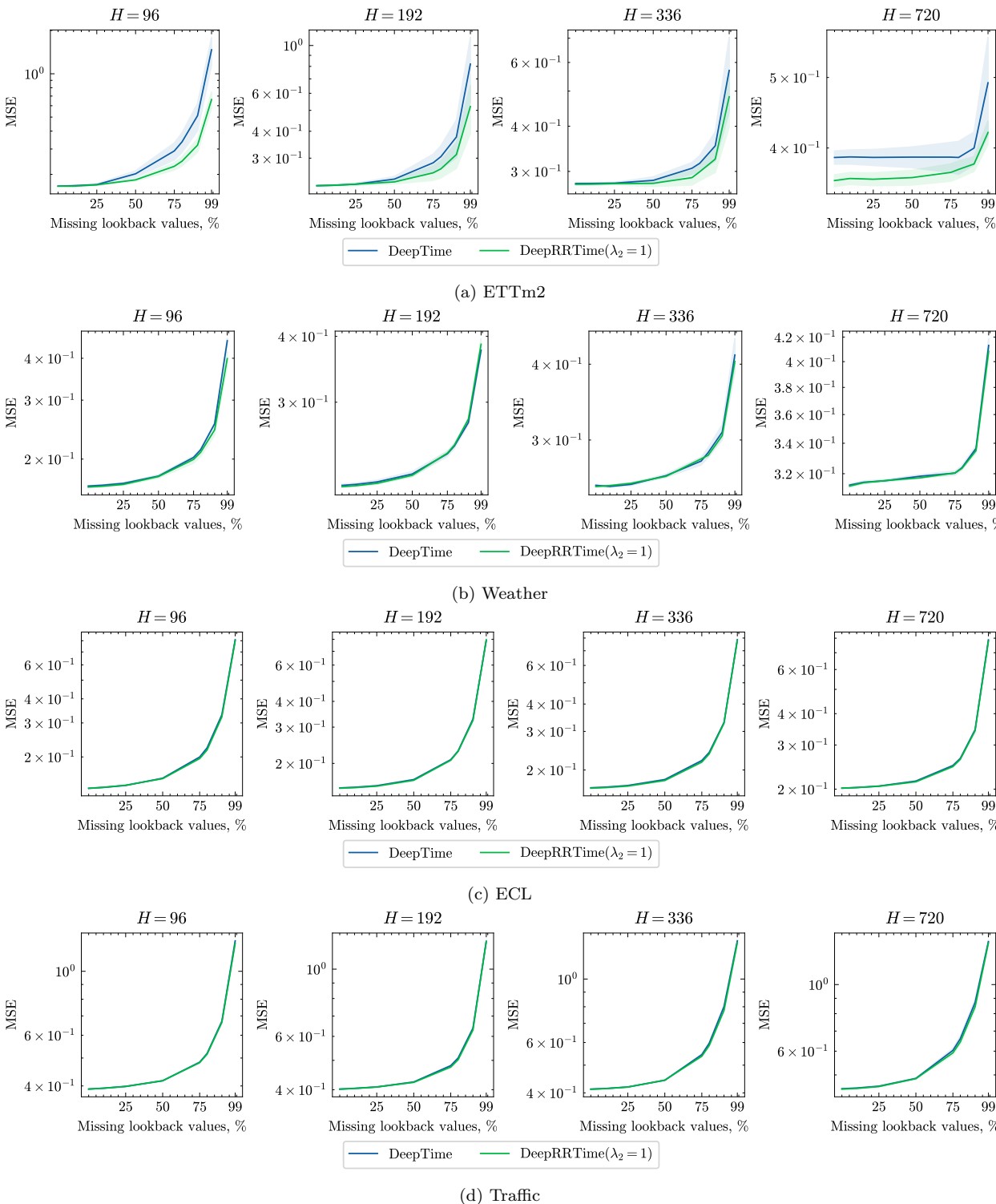

Figure 8: Plots of mean squared error (MSE) of DeepRRTime and DeepTime as a function of missing lookback values percentage for different forecast horizons on ETTm2 (top), Weather (middle) and ECL (bottom) dataset. The shadow areas report standard deviations over 10 network initializations. For the ETTm2 dataset, the performance of DeepTime deteriorates more significantly for higher missing rates than our model. For the bigger datasets (i.e., Weather, ECL, and Traffic), the performances of DeepRRTime and DeepTime are very similar. We skip this experiment for ILI due to the shorter forecast horizons used for this dataset.

Figure 9: Forecasting Plots on Exchange: We visually compare between DeepTime, DeepRRTime and PatchTST. The 8 rows of a column denote the 8 different time-variates and all plots within the same column are taken from the same time-period. The sample-IDs for the visualisation are randomly sampled and indicated at the top of each column along with the mean-squared error (MSE) of forecasts for each method. More specifically, the MSE at the top of each column is the MSE averaged over all 8 plots in that column. In several cases, we observe that the DeepRRTime are

