# OpenReview forum: "DeepRRTime: Robust Time-series Forecasting with a Regularized INR Basis"
_TMLR — Accepted by TMLR_

### Review · Reviewer_BCmj · 2024-10-21

**Summary Of Contributions:**

The paper presents DeepRRTime, a novel extension to the DeepTime framework for time-series forecasting, incorporating a covariance regularization technique.
1. The authors propose a simple, computationally inexpensive regularizer that encourages the time-indexed basis learned by the model to be more unit-standardized and less mutually correlated. This regularization is designed to improve the conditioning of the ridge regression step, leading to more robust forecasts.
2. DeepRRTime is evaluated on six real-world multivariate time-series benchmarks and is shown to outperform the original DeepTime and other time-index models. It demonstrates state-of-the-art results, especially on the challenging Exchange dataset, where previous models struggled due to its low signal-to-noise ratio and non-stationarity.
3. DeepRRTime remains robust even when up to 90% of the lookback data is missing, outperforming DeepTime and other models.
4. The regularizer helps maintain generalization when training data is limited, reducing the performance gap between reduced and full datasets.

**Audience:**

Yes

**Claims And Evidence:**

Yes

**Requested Changes:**

see Weaknesses

**Strengths And Weaknesses:**

Strong Aspects

1. The introduction of a simple yet effective regularization term to improve the robustness of ridge regression is a strong innovation. The approach is theoretically grounded and shows clear benefits in practical applications.
2. The submission provides comprehensive experimental results across six real-world datasets, demonstrating the effectiveness of DeepRRTime. The improvements in performance over both DeepTime and other time-index models are consistent and significant, particularly on the challenging Exchange dataset.
3. The robustness tests, including scenarios with missing data, reduced training datasets, and test-time interpolation, add depth to the evaluation and illustrate the versatility of the proposed method.
4. The regularization technique is easy to implement and integrates smoothly with the existing DeepTime framework. This practicality is a valuable asset for deploying the model in real-world settings, where complex modifications can hinder adoption.
5. The work is theoretically well-supported, leveraging insights from linear regression theory to justify the regularization. This grounding in theory enhances the credibility of the approach and explains why the improvements are observed.

Weaker Elements

1. The paper acknowledges that hyperparameter tuning was not extensively conducted, which might mean the reported performance does not fully capture the potential of DeepRRTime. Future work could focus on a more thorough exploration of parameters like the regularization coefficient $\lambda_2$ and the lookback multiplier $\mu$, perhaps integrating automated hyperparameter tuning methods to optimize performance further.
2. While the six datasets used are diverse, the evaluation could benefit from additional datasets that exhibit different characteristics (e.g., datasets with higher frequency seasonal patterns, more non-stationary behaviors, or data from different application domains like finance or healthcare). Expanding the range of datasets would strengthen the claims of robustness and generalizability.
3. The reliance on manually tuned parameters, such as $\lambda_2$ and $\mu$, could limit the scalability of the model. Automated techniques for dynamic parameter adjustment during training might improve usability in scenarios where these parameters are hard to determine or need to adapt based on the data.
4. Although the regularization is described as computationally inexpensive, the need for fine-tuning across multiple datasets could introduce additional computational overhead, particularly when large-scale datasets or complex multivariate settings are involved. The authors might consider discussing strategies to mitigate such overheads, such as batch optimization or efficient regularization scheduling.

---

> ### Author Response · Authors · 2024-11-15
> **Review Response.**
>
> We thank the reviewer for their review, highlighting both the strengths of our work and constructive areas for improvement. We agree that addressing the suggested weaker elements presents valuable directions for future research. In response, we have incorporated several of these recommendations into the future work section of our paper. Specifically, we have added a future work item focused on exploring more comprehensive hyperparameter tuning, including automated tuning methods to optimize parameters such as the regularization coefficient and lookback multiplier. Additionally, we recognize the potential value of evaluating DeepRRTime on datasets with diverse characteristics, such as high-frequency seasonal patterns or pronounced non-stationarity, to further explore its robustness and generalizability, and include this item in future work. We hope this addresses all your outstanding concerns, and we look forward to resolving any additional questions during the remaining discussion period.

---

> > ### Comment · Reviewer_BCmj · 2024-12-10
> > **Reaction to Rebuttal**
> >
> > While the planned future work is commendable, I believe the following points could further strengthen the current version of the paper:
> >
> > The paper could benefit from preliminary explorations related to the proposed future work. For example:
> > Conduct a sensitivity analysis of hyperparameters on the current datasets, even at a small scale, to showcase their potential impact on performance.
> > Introduce one or two additional datasets (e.g., high-frequency or non-stationary) to provide a glimpse into the robustness of the proposed method.
> > Additional Discussion:
> >
> > The response mentions computational overhead and scalability concerns due to manual tuning. Including concrete suggestions or strategies in the paper, such as reducing the number of tunable parameters or employing efficient optimization methods, would be valuable.

---

> > > ### Author Response · Authors · 2025-01-30
> > > **Thank you for your response!**
> > >
> > > We thank you for your careful consideration of our rebuttal and addtional recommendations. We address them as follows:
> > >
> > > * **Analysis of Hyperparameters:** We point out that our paper already includes an analysis of hyperparameters when training with small datasets. Figure 3 visualises the forecasting-error for varying values of regularization strength ($\lambda_2$): we generally observe that increased regularization strength results in an increased performance (lower forecasting error). We note that we also identified the optimal lookback-multiplier $\mu$ for each value of $\lambda_2$. Due to limited computational budget, we do not optimize the $\lambda_2$ hyperparameter when utilizing the full training-data and leave this for future work.
> > > * **Additional Datasets for robustness analysis:** We agree that it is valuable to conduct a targeted analysis of robustness by considering multiple-datasets sharing similar attributes such as seasonality and stationarity. Since this requires separate research efforts to collect and categorize datasets, we leave this as future work.
> > > * **Efficient Hyperparameter Optimization:** We could reduce the tunable parameters by learning optimal $\lambda_2$ using a gradient-based optimization instead of manual tuning. While this idea is interesting, it would require dedicated research-efforts and literature-survey and we have included this in the future-work section.
> > >
> > > We thank you again for your thorough review and firmly believe that your suggestions have helped enhance our paper.

---

### Review · Reviewer_ZUzk · 2024-10-28

**Summary Of Contributions:**

In this paper, the authors build on the DeepTime model. DeepTime works by having
a neural network to infer hidden predictions at arbitrary historic and future
time-points and then apply ridge-regression on the hidden-representations of the
lookback window. Then,  the neural network and the ridge regression top head can
be used to forecast for the future time-points.

The particular extension of this paper lies in a regularization term. The
authors propose to not only fit the ridge regression on the forecasting task,
but also on de-corellating the hidden representations. In detail, they add an
auxillary loss that pushes the centralized covariance matrix of the hidden
representations towards the identity matrix.

The authors evaluate their method DeepRRTime on "standard" time-series
forecasting, forecasting with missing values, in the small training data regime
and in a setting where the frequency in the test data differs from the training
data.

**Audience:**

Yes

**Claims And Evidence:**

No

**Requested Changes:**

# Important:

- Present empirical evidence, that your methods is indeed always very fast
- I would like to see standard deviations for Table 2, so basically Figure 2 also for the other datasets, to see if your method is really superior when dealing with missing values.
- Why should I use Time-Indices at all over, say, PatchTST? Can you compare it against PatchTST (or another regular sequential model) on the 50% missing value setting? in PatchTST, you could for example simply 0 the missing values or linearly interpolate them first. Or you compare against that on the 10% training data setting. Just an indication, that I should use Time-Indices at all in some settings.

# Minor:

- I would move theorem 2 into the main text, somewhere around section 4.2. When I first read the main text, I had the issue that 4 tells me "We want to maximize the smallest and minimize the largest eigenvalue." And then at the beginning of 4.2 you say "We do pushing G_theta against the Identity instead because it worked better." I was confused and asking myself why then even talk about the maximization and minimization of the eigenvalues at all? However, Theorem 2 and the discussion around it give the connection to understand that. Thus, they are crucial and should go into the main text. "Similarly, the
largest eigenvalue of Gθ also cannot be very large when LCov(θ) is small." Can you elaborate on that in detail, as done in (10)-(12)?
- I do not like calling the Z matrix the "basis". A basis of R^d is a very well-established term for a sequence of vectors in R^d, and its not how you use it here. I especially stumbled upon the first sentence in section 3.1 with respect to this regard. The part "Z \in R^D represents the values of the D-dimensional base" sounds just wrong as a D-dimensional basis (in the sense of basis as always used in linear-algebra) would need D different vectors.

**Strengths And Weaknesses:**

# Strengths and Weaknesses
+ Paper and method is easy to understand
+ The regularization term is well motivated and its usefulness is intuitive
+ The experiments are extensive: Normal setting, low data regime, different test frequencies etc

- The results are to me not really promising: In Table 6, the enhancement do not look that large between DeepTime and DeepRRTime
- DeepTime and DeepRRTime are most of the time beaten by PatchTST in Table1. Why use Time-Indices at all?
- The computationally inexpensiveness has to be showed somehow. I mean, for Equation (4) you would have to compute ZZ^T which is naively L^2*d and then also solve the equation system

# Are all Claims Proven?
- To some extent: Looking at Table 6: That DeepRRTime beats normal DeepTime,especially for the smaller horizons, may not be significant as the stdvs indicate
- Same problem I have with Table 2, the 50% missing experiment
- In Table 4 (and the picture below) it is more convincing.

As I have to make a binary decision, I would, at the current stage, slightly lean towards a "No", but I am willing to change my decision depending on the outcome of my requested changes

# How interesting is it to the Audience?

- I would say it is somehow interesting. It is an easy to implement
  regularization term which seems to help (to a slight extent / in the low
  training-data regime). So especially when having low amount of training data,
  one may would try to test DeepRRTime. However that Deep(RR)Time seems not be
  fully competitive with PatchTST limits the importance.

---

> ### Author Response · Authors · 2024-11-15
> **Review Response**
>
> We thank the reviewer for their time in reviewing our submission, appreciating the strengths and providing insightful recommendations to improve the paper. We address the weaknesses and answer your questions in the following:
>
> **[W1] Computational and memory cost**
>
> We thank the reviewer for this suggestion and have added Table 11 where we provide a comparison between PatchTST and DeepRRTime in terms of peak memory usage and wall-clock time per epoch during both training and inference, using a single NVIDIA Tesla V GPU (16GB). Our results show that DeepRRTime is consistently faster than PatchTST in both training and evaluation. While DeepRRTime is generally more memory-efficient, some exceptions were observed (e.g., Exchange/720). Notably, DeepRRTime did not encounter any out-of-memory (OOM) issues across all forecast horizons on this GPU, whereas PatchTST experienced OOM errors for all horizons on the Traffic dataset. These findings highlight the computational efficiency of our model DeepRRTime compared to historical-value models such as PatchTST.
>
> ***
>
> **[W2] Improvement of performance compared to DeepTime and its statistical significance**
>
> Thank you for bringing up the question of statistical significance. We agree that it is crucial to factor this into any interpretation of the results. Indeed, it is for this reason that we included the standard deviations of errors in our appendices. Please note that our original submission includes the standard deviations for Table 1 in Table 6, and the standard deviations for Table 2 in Table 9. We apologize for having failed to note clearly in Sec. 5 the main text that Table 9 in the appendix contained the standard deviations corresponding to Table 2, and have adjusted the text accordingly.
>
> Before directly answering the question with statistical significance results, we wish to provide additional context for interpreting the results. Our main contribution is a new regularization technique, and as you acknowledged above and we discussed in Sec. 5.4 of the manuscript, the improvements of improved regularization may naturally be more evident when dealing with smaller training datasets. Thus, we find it all the more noteworthy that several improvements of DeepRRTime over DeepTime are statistically significant even when the full dataset is available (i.e., in the regular forecasting and missing-value forecasting settings). Furthermore, the number of statistically significant improvements substantially increase in the missing-value setting as compared to the regular forecasting. We find this particularly interesting because the missing-value test is conducted using the exact same trained models tested in the regular forecasting setting. We feel that this demonstrates that the challenging missing-value test, designed to connect the specific advantages conferred by our regularizer to a test setting directly relevant to real-world robustness, can precisely isolate and confirm the expected benefits of our regularization method.
>
> Prompted by your comment, we conducted a more rigorous assessment of the significance based on Welch's t-test. The one-sided p-values for the mean errors across seeds are reported below; the null hypothesis here is that each error result observed for DeepRRTime is not better than the corresponding error for DeepTime, and the p-value is the probability that the observed level of difference was observed despite this being the case.

---

> ### Author Response · Authors · 2024-11-15
> **Review Response**
>
> |   |   | Regular  Forecasting |  | Missing-Value Forecasting |  |
> |---|---|---|---|---|---|
> |   |   | MSE | MAE | MSE | MAE |
> | ETTm2 | 96 | 0.31220363 | 0.661585414 | **0.00138855** | **0.00386958** |
> |   | 192 | 0.9169776 | 0.812845487 | 0.13715673 | 0.18141168 |
> |   | 336 | 0.22920168 | 0.315992682 | 0.08389574 | 0.08418046 |
> |   | 720 | **9.48E-06** | **8.30E-06** | **6.32E-06** | **6.49E-07** |
> | ECL | 96 | **0.0002448** | **0.001132138** | **0.01014307** | **7.05E-05** |
> |   | 192 | **0.00113214** | **0.000103712** | **0.00217865** | **0.00059352** |
> |   | 336 | 0.10113955 | **0.01536049** | **1.81E-07** | **8.88E-08** |
> |   | 720 | _0.96751023_ | 0.839961141 | **0.02886595** | **0.02714545** |
> | Exchange | 96 | 0.05333092 | 0.070524199 | **0.00144792** | **0.00058923** |
> |   | 192 | 0.77062022 | 0.099436944 | **0.02245797** | **0.00271907** |
> |   | 336 | **1.81E-05** | **1.07E-05** | **0.0003375** | **0.00045043** |
> |   | 720 | 0.06620207 | 0.082152599 | **0.02502503** | **0.03157751** |
> | Traffic | 96 | 0.07195579 | **0.005968114** | 0.24766801 | **6.16E-05** |
> |   | 192 | 0.84856516 | 0.5 | **1.81E-07** | **3.00E-09** |
> |   | 336 | 0.6029121 | **0.000361852** | **6.14E-09** | **3.97E-12** |
> |   | 720 | 0.19046095 | 0.163563756 | 0.05395065 | 0.05202705 |
> | Weather | 96 | 0.10113955 | 0.057972227 | _0.97151735_ | 0.7978649 |
> |   | 192 | 0.78247412 | 0.5 | _0.98911354_ | _0.96927951_ |
> |   | 336 | **0.01316985** | **0.005303186** | 0.0730186 | **0.04275086** |
> |   | 720 | **0.00571167** | **0.003520983** | **0.00710047** | **0.00485499** |
> | ILI | 24 | **0.00078261** | **0.001272104** | **0.00408419** | **0.0155755** |
> |   | 36 | 0.40539609 | 0.127618138 | 0.34288425 | 0.20665302 |
> |   | 48 | 0.44625033 | 0.632202635 | 0.45255198 | 0.65726215 |
> |   | 60 | 0.55642524 | 0.605079885 | 0.59371749 | 0.64287789 |
>
> This table shows the p-values comparing between DeepTime and DeepRRTime based on one-sided Welch's t-test: statistical significant cases of improvement are bolded while statistical significant cases of degradation are italicized.
>
> Previously, we only judged a result significant by the criterion that the difference in mean errors exceeded 3 times the combined standard error. With this more precise assessment of the uncertainty in the measurements, the number of statistically significant results (at the typical threshold of 0.05) has increased from 13 to 17 for the standard forecasting results in Table 1 and from 21 to 30 for the missing-value setting results in Table 2. We also note several more p-values just above the 0.05 significance level, and well below 0.1. Thus, we reaffirm our confidence that the improvements due to our regularizer are indeed statistically significant. We have adjusted the discussion of Table 1 accordingly.
>
> Although we approached the test with the null hypothesis that DeepRRTime is not better than DeepTime (in mean error across seeds), we do incidentally observe a couple cases where the p-value is unusually high. This could be an indication of DeepRRTime actually underperforming DeepTime to a statistically significant extent.
> * For the p-values corresponding to standard forecasting in Table 1, one of the 40 reported values exceeds 0.95. We feel this is consistent with the expected behavior of p-values even if the true mean error of DeepRRTime is not worse than that of DeepTime. By definition of the p-value, we expect a reasonable chance of observing such a result by chance given this many tests.
> * On the other hand, for the p-values corresponding to missing-value forecasting results reported in Table 2, we note that three of four p-values corresponding to the two shortest horizons of Weather exceed 0.95. We feel this may be an indication that DeepRRTime truly underperforms DeepTime to a statistically significant extent on those tests. We had already acknowledged in the original submission that "DeepRRTime is more resilient to this test than DeepTime on all datasets except Weather, where they perform comparably". We feel this is still consistent with the new significance analysis; the absolute magnitude of underperformance is small (they differ from one another by <1% in the missing-value test, compared to the 5% degradation relative to the original forecasting test). Thus, we do not feel this result changes our overall claim that DeepRRTime is uniformly better than or comparable to DeepTime. Still, we have amended our original statement to "DeepRRTime is more resilient to this test than DeepTime on all datasets except Weather; on the shortest two horizons of Weather, DeepRRTime underperforms DeepTime by a statistically significant but nonetheless small amount."

---

> > ### Author Response · Authors · 2024-11-15
> > **Review response**
> >
> > In summary, we remain confident in our claims after a closer inspection of the standard deviations reported in the original manuscript. It appears that a few more results are statistically significant than we had originally realized. The analysis confirms that a case of minor underperformance we previously identified is indeed significant, and we have acknowledged this in the text. Overall, the results are consistent with our claim that DeepRRTime is better than or comparable to DeepTime across all our tests, and that DeepRRTime is generally much more resilient to the missing-values setting than DeepTime.
> >
> > Additionally, based on your feedback, we have extended the experiment in Fig. 2 to other datasets and include these in Figure 8 in the appendix. Please note that only partial results are included currently — we will add the plots for the remaining datasets/horizons/seeds in the final revision. Previously, we simply fixed the lookback-multiplier=1 in Figure 2 as a controlled experiment to highlight the effectiveness of the regularization. To make the connection between Table 2 and Fig. 2 more direct, we have recomputed Fig. 2 using optimal values of the lookback multiplier. Accordingly, all plots in Fig. 8 are also based on optimal lookback multipliers. As before, we fix the regularization-strength $\lambda_2=1$, although as noted elsewhere tuning of this parameter could further enhance the results. These plots illustrate the improvement of DeepRRTime over DeepTime as a function of the percentage of missing values in the lookback window. Overall, these visualizations indicate that DeepRRTime consistently outperforms (e.g., Exchange/ETTm2) or matches (e.g., Weather/ECL) DeepTime in the missing lookback value settings.
> >
> > To conclude, we would like to emphasize the simplicity of our regularizer and the consistent improvements it brings when applied to DeepTime, particularly when the model is tested in more challenging scenarios such as smaller training dataset sizes, missing values in the lookback window, and different forecast frequencies at train and test time.
> > ***

---

> > > ### Author Response · Authors · 2024-11-15
> > > **Review Response**
> > >
> > > **[W3] Benefits of time-index models compared to historical-value models**
> > >
> > > We would like to emphasize that deep time-index models for time series forecasting are a relatively new family of models, while historical-value models have benefited from extensive research over many years. Our work introduces a key methodological refinement, making time-index models more competitive with historical-value models and paving the way for further advancements in this area.
> > >
> > > Moreover, our study highlights several inherent advantages of time-index models that make them particularly suitable for challenging real-world scenarios. Specifically:
> > > * **Handling missing values:** Time-index models can natively handle missing values in the lookback window without requiring imputation, providing an advantage over historical-value models in such settings. Based on your suggestion, we evaluated PatchTST on the ETTm2 dataset with missing values in the lookback window and reported the results in Table 12. We explored two approaches to handle missing values: (i) replacing missing values with zeros and (ii) applying linear interpolation. The results indicate that DeepRRTime demonstrates significantly greater robustness to missing values compared to PatchTST under these conditions. While we acknowledge that an imputation strategy tailored to a specific dataset and model could yield improved results, we would like to emphasize that one of key advantages of time-index models is their inherent ability to handle missing values natively.
> > >
> > > * **Adaptability to different frequencies:** Time-index models can naturally accommodate different frequencies between training and testing phases, allowing them to adapt flexibly to varying data sampling rates.
> > >
> > > These capabilities illustrate the potential of time-index models to address complex forecasting tasks in challenging settings that are common in real-world applications, suggesting promising directions for further research and practical deployment
> > >
> > > Moreover, as demonstrated in the newly added Table 11 of the Appendix, deep time-index models are generally more computationally efficient than PatchTST—sometimes dramatically so. This alone could be a reason to prefer these models in certain practical settings; DeepRRTime is generally second to PatchTST among the many baselines we compare against (see Table 5), positioning it as a worthwhile candidate for time-constrained applications desiring near-SOTA performance.
> > >
> > > Additionally, we note that DeepRRTime outperforms PatchTST on the Exchange dataset, which is known to be challenging due to its low signal-to-noise ratio and high non-stationarity. This suggests that the design of deep time-index models may render them more suited to problems that are characteristically similar to Exchange. We hope that our response resolves your concern about advantages of time-index models — if not, we request you to kindly share your points of concern and we would be very happy to address them.
> > >
> > > ***
> > >
> > > **[W4] Moving Theorem 2 into the main text**
> > >
> > > We thank the reviewer for this suggestion and have moved Theorem 2 to Section 4.2 of the submission.
> > >
> > > ***
> > >
> > > **[W5] Discussion about the "basis" term**
> > >
> > > Thank you for raising this concern with our terminology. After some reflection, we agree with your concern.
> > >
> > > First, we wish to clarify our meaning with phrasing such as "$z_τ \in \mathcal{R}^D$ represents the values of the D-dimensional basis at time τ". We view each output node of the INR as a function of the time-index tau. Our meaning here is that the set of D output nodes, viewed as D functions of tau, form a spanning subspace of a set of functions of time. Thus, the vector space for which we meant them to be a basis was the space of functions of time, not $R^D$. Rather, we meant that the D-dimensional function of time given by the set of these functions has the value $z_\tau$ at the particular time τ.
> > >
> > > Nonetheless, the term ‘basis’ is still not generally applicable from this perspective. Except in the special case where the D functions of time are orthogonal to one another (with respect to the inner product over tau), they will generally not form a basis of the subspace that they span because they will not be linearly independent. Although achieving this linear independence is ultimately the motivation for our regularizer, we no longer feel the term is appropriate given that the set cannot be guaranteed to always constitute a basis.
> > >
> > > Rather, we feel that the correct term for our meaning is a ‘frame’, rather than a ‘basis’. This term refers to an overcomplete set of spanning vectors. We propose to use it as a drop-in replacement for the term basis throughout the paper. Because this term may be less familiar to some of the target readership, we propose to also include a short definition in the introduction. If you agree with this recommendation, we will introduce these changes in the final manuscript.

---

> > > > ### Comment · Reviewer_ZUzk · 2024-11-27
> > > > **Reaction to Rebuttal**
> > > >
> > > > Dear Authors,
> > > > thank you for your extensive rebuttal. I am now convinced that DeepTime models are important to study in general. However, I am still not fully sure about the importance of your particular extension:
> > > > - In your table above, your regularization only improves performance significantly in 46/96 cases
> > > > - I however do see the point that it is indeed useful for the small training data regime.

---

> > > > > ### Author Response · Authors · 2024-11-28
> > > > > **Thank You!**
> > > > >
> > > > > We thank you for your careful consideration of our rebuttal and giving us an opportunity to offer further clarifications.
> > > > >
> > > > > Firstly, we emphasize that, in the table above, the same models (i.e., same model parameters) are evaluated in both the regular and missing-value settings. The regular forecasting error is significantly improved by our regularizer on at least one error metric in 10/24 settings, and is never significantly worse. Interestingly, the exact same models tested in the missing-value setting show substantial improvements in 16/24 settings. Together, we take these results as evidence that the DeepRRTime solutions are more robust. Our many experiments indicate that the improvement in robustness simultaneously confers advantage across a diverse set of perturbed conditions, including training with less data, inference with missing values, and inference at a different frequency than training.
> > > > >
> > > > >
> > > > > Additionally, the theoretical insights presented in this paper lays the foundation for future work to improve the design of deep time-index models. Covariance regularization is just one simple approach to build upon the underlying theory — i.e., improving the conditioning of the deep time-index representations is a useful inductive bias for making robust deep time-index models. Future work may build upon these insights to propose better regularization objectives, or to otherwise refine the deep time-index model paradigm.
> > > > >
> > > > >
> > > > > Overall, DeepRRTime has no substantial downsides compared to regular DeepTime (implementation is simple and additional computational cost is small), sometimes significantly improves performance on regular forecasting benchmarks, and substantially increases robustness to a wide set of perturbations. The perturbations we studied were specifically designed to reflect circumstances that can arise for real-world time series forecasting applications, and where non-robust solutions may be expected to break down. Moreover, we feel that its improved robustness may make DeepRRTime a more stable foundation than the original DeepTime for those interested in exploring future variants. **We imagine that this simple but powerful refinement should become a standard addition to the toolkit for any researchers interested in further expanding the class of deep time-index models.** As this new class of models certainly merits further research, we hope you will agree that it would be worthwhile to share our developments with your readership.

---

### Review · Reviewer_V2ZN · 2024-10-30

**Summary Of Contributions:**

This paper addresses the challenge of time-indexed models for time series forecasting and introduces DeepRRTime, an enhanced version of the existing DeepTime model. DeepTime leverages Implicit Neural Representations (INRs) as a latent basis, which are then combined with a Ridge regressor through a meta-learning optimization. This combination enables the model to adapt to different look-back windows effectively.

DeepRRTime improves upon DeepTime by incorporating covariance regularization over the INR basis representation (before the ridge regressor). Specifically, it adds a term to the loss function that encourages the off-diagonal elements of the covariance matrix to approach zero while pushing the diagonal elements toward one. The rationale is that Ridge regression can be more robust when operating on a basis of linearly uncorrelated vectors with unit norms.

Experiments conducted on standard LTSF datasets demonstrate that DeepRRTime outperforms the original framework in scenarios with limited sample sizes and when the look-back window contains missing data. In traditional settings, where such challenges are absent, both methods exhibit comparable performance.

**Audience:**

Yes

**Claims And Evidence:**

No

**Requested Changes:**

Please see the section on weaknesses. I believe that the suggested improvements could significantly enhance the quality of the paper.

**Strengths And Weaknesses:**

### Strengths

S1. The paper is clearly written and easy to follow, with a well-defined scope of contributions. Figure 1, in particular, is an excellent visual aid that enhances understanding. Moreover, the advantages of time-indexed models over traditional historical-value models for time series are clearly articulated and well-supported.

S2. Although the improvement over DeepTime may seem incremental at first glance, the focus on scenarios with limited number of samples and missing values in the look-back window address challenges frequently encountered in real-world applications. The experiments effectively validate the proposed regularization approach, reinforcing the claims made in the introduction and demonstrating its practical relevance.


### Weaknesses

W1. The authors compare their approach to Gaussian and TimeFlow in Table 1, but I would have liked to see these methods included in the comparisons for Table 2 and Table 3 as well. If I am not mistaken, both of these time-index models are also capable of handling forecasting tasks with missing values in the look-back window, so a broader comparison would be valuable for evaluating performance under these conditions.

W2. It would be beneficial to include some forecasting plots in the appendix, such as comparisons between DeepRRTime, DeepTime, GP, TimeFlow, and PatchTST. Visualization of these results would provide a clearer understanding of how the models perform in practice.

W3. The inclusion of the Martingale baseline is nice to illustrate the complexity of the forecasting problem. However, it would be helpful to also include a simple repeat baseline where the look-back window is repeated over the forecast horizon. For example, for a horizon of 336, simply repeat the last 336 values. Adding this baseline could provide valuable additional insight into the difficulty of the forecasting task.

---

> ### Author Response · Authors · 2024-11-15
> **Review Response**
>
> We thank the reviewer for their time in reviewing our submission, appreciating the strengths and providing insightful recommendations to improve the paper. We address the weaknesses and answer your questions in the following:
>
> **[W1] Comparison with other time-index models with missing lookback values**
>
> We thank the reviewer for this valuable suggestion. Our results in Table 2, which compare models in the standard multivariate forecast setting, indicate that TimeFlow and GP models are not competitive with DeepTime and DeepRRTime models. Additionally, as detailed in Appendix C, we encountered methodological limitations when testing TimeFlow, resulting in irreproducible outcomes. Consequently, we focused on comparisons between DeepTime and DeepRRTime, which showed more reliable and robust performance across various settings. For completeness, we have evaluated TimeFlow on the ETTm2 dataset in the missing-value setting and present the results below:
>
> |  | TimeFlow |  | DeepRRTime |  |
> |---|---|---|---|---|
> | H | MSE | MAE | MSE | MAE |
> | 96 | 0.2913±0.0230 | 0.3409±0.0088 | **0.1835**±0.0047 | **0.2841**±0.0063 |
> | 192 | 0.3986±0.0221 | 0.4066±0.0091 | **0.2303**±0.0048 | **0.3120**±0.0060 |
> | 336 | 0.5416±0.0208 | 0.4824±0.0142 | **0.2787**±0.0044 | **0.3434**±0.0054 |
> | 720 | 0.6709±0.0035 | 0.5624±0.0032 | **0.3634**±0.0028 | **0.3942**±0.0042 |
>
> As shown, DeepRRTime significantly outperforms TimeFlow across all forecast horizons in terms of both MSE and MAE. This further highlights the robustness of DeepRRTime in handling missing values and its effectiveness compared to existing time-index models under challenging conditions.
>
> ***
>
> **[W2] Visualisations of model predictions**
>
> We added visualizations comparing the forecasts of DeepTime, DeepRRTime and PatchTST in Figure 9 (in appendix) on the Exchange dataset. We selected five random examples from the test set (corresponding to the five columns of the figure), and visualized the predictions separately for each column of the dataset (corresponding to the eight rows of the figure).  Interpreting these plots together with the results in Table 1, we can make the following observations: i) PatchTST clearly underperforms the time-index models on this challenging dataset both qualitatively and quantitatively; and ii) the qualitative effect of the regularization term on the DeepRRTime forecasts as compared to DeepTime is also clearly visible: for instance the DeepTime forecasts exhibit far more variance than those of DeepRRTime.
>
> ***
>
> **[W3] Repeat baseline**
>
> We thank the reviewer for this suggestion and have evaluated the repeat baseline alongside the martingale baseline, another trivial approach included in the paper. Our findings indicate that both baselines consistently underperform relative to DeepRRTime on all the metrics except Exchange/96 where the Martingale baseline is marginally better in terms of the MAE. However, the martingale baseline demonstrates competitive performance with state-of-the-art models on the Exchange dataset, which motivated our decision to feature it in the main paper. In particular, the martingale baseline outperforms PatchTST (see Table 1), whereas DeepTime and DeepRRTime outperform the martingale. Since the martingale has essentially no predictive value, we consider this a key result to emphasize (i.e., PatchTST and the other models are worse than nothing on Exchange, whereas DeepTime and DeepRRTime are significantly better). Moreover, the PatchTST authors explicitly acknowledge the Exchange dataset as difficult and opted to exclude it from their paper, so our comparison of PatchTST against the martingale on this dataset is a missing perspective on this otherwise state-of-the-art time series model.
>
> Below, we provide a table comparing the repeat baseline, martingale baseline, and our proposed method. As shown, the repeat baseline does not perform competitively with the other approaches presented in Table 1, further justifying our choice to emphasize the martingale baseline due to its relative effectiveness on challenging datasets like Exchange.

---

> > ### Author Response · Authors · 2024-11-15
> >
> > |  |  | Repeat |  | Martingale |  | DeepRRTime |  |
> > |---|---|---|---|---|---|---|---|
> > |  | H | MSE | MAE | MSE | MAE | MSE | MAE |
> > | Exchange | 96 | 0.1640 | 0.3009 | 0.0811 | **0.1964** | **0.078** | 0.197 |
> > |  | 192 | 0.3577 | 0.4514 | 0.1671 | 0.2887 | **0.153** | **0.284** |
> > |  | 336 | 0.6615 | 0.6259 | 0.3057 | 0.3978 | **0.257** | **0.375** |
> > |  | 720 | 1.5890 | 0.9760 | 0.8101 | 0.6764 | **0.541** | **0.54** |
> > | ECL | 96 | 0.3653 | 0.3570 | 1.5878 | 0.9455 | **0.137** | **0.238** |
> > |  | 192 | 0.3171 | 0.3409 | 1.5962 | 0.9507 | **0.152** | **0.251** |
> > |  | 336 | 0.2476 | 0.3129 | 1.6178 | 0.9613 | **0.165** | **0.267** |
> > |  | 720 | 0.4693 | 0.4393 | 1.6468 | 0.9754 | **0.202** | **0.303** |
> > | ETTm2 | 96 | 0.2631 | 0.3005 | 0.2665 | 0.3278 | **0.166** | **0.258** |
> > |  | 192 | 0.3763 | 0.3713 | 0.3400 | 0.3707 | **0.224** | **0.3** |
> > |  | 336 | 0.5808 | 0.5095 | 0.4121 | 0.4099 | **0.276** | **0.338** |
> > |  | 720 | 0.7166 | 0.5681 | 0.5216 | 0.4656 | **0.368** | **0.397** |
> > | Traffic | 96 | 1.4388 | 0.5629 | 2.7145 | 1.0772 | **0.39** | **0.274** |
> > |  | 192 | 0.9387 | 0.4177 | 2.7471 | 1.0851 | **0.402** | **0.278** |
> > |  | 336 | 0.6520 | 0.3110 | 2.7885 | 1.0941 | **0.416** | **0.285** |
> > |  | 720 | 1.4111 | 0.5643 | 2.8096 | 1.0968 | **0.45** | **0.307** |
> > | Weather | 96 | 0.3490 | 0.3329 | 0.2591 | 0.2542 | **0.166** | **0.222** |
> > |  | 192 | 0.4443 | 0.3951 | 0.3092 | 0.2917 | **0.207** | **0.26** |
> > |  | 336 | 0.5169 | 0.4305 | 0.3764 | 0.3377 | **0.251** | **0.298** |
> > |  | 720 | 0.5298 | 0.4272 | 0.4652 | 0.3935 | **0.312** | **0.348** |
> > | ILI | 24 | 10.554 | 2.392 | 6.213 | 1.622 | **2.317** | **1.044** |
> > |  | 36 | 8.443 | 2.131 | 7.714 | 1.906 | **2.253** | **1.022** |
> > |  | 48 | 3.131 | 1.144 | 7.851 | 1.952 | **2.292** | **1.033** |
> > |  | 60 | 4.589 | 1.507 | 6.885 | 1.788 | **2.301** | **1.035** |
> >
> > ***
> >
> > We hope this response addresses all your outstanding concerns, and we look forward to resolving any additional questions during the remaining discussion period.

---

### Review · Reviewer_kRWA · 2024-10-31

**Summary Of Contributions:**

This work focuses on **time-index models for time series forecasting**, which have recently shown quite competitive performance compared with the common historical-value models. More specifically, the authors propose a **regularization term** for the recently proposed DeepTime framework, whose time-index basis is built upon an implicit neural representation, and a meta-learning module generates predictions with ridge regression. The proposed regularization term forces the elements to be less mutually correlated *to boost the robustness of the ridge regression part*. Experimental evaluation on forecasting with benchmark datasets, especially with *synthetically generated missing values* in the loopback windows, shows improved performance for the proposed method compared to baselines.

**Audience:**

Yes

**Claims And Evidence:**

No

**Requested Changes:**

The following aspects could significantly improve the quality of this work and the impact of the contribution:
1. Based on **[W1]**, the authors should better justify the impact of the technical contribution beyond the chosen DeepTime architecture. They should consider updating the part of related work with some references to relevant topics studied in ridge regression to address **[W2]**.
2. Authors should consider adding additional baselines for historical-value methods **[W3]** and for the experiment with missing points **[W4, W5]** to enable more fair performance comparisons. Based on **[W6]**, since the experimental evaluation (e.g., in the number of runs) followed by the authors differs from other baselines, they should evaluate performance with the same protocol for all methods.
3. It is important that the computational cost of the meta-learning framework is captured along with other time/memory cost comparisons with the baselines **[W7]**. Plots on the predictions for the several setups could further enable qualitative comparisons of methods **[W8]**.

**Strengths And Weaknesses:**

The following significant strong aspects of this study are highlighted:
1. The authors showcase in several experiments the **potential of time-index models as an alternative to historical-value models** that are often prone to overfitting/poor generalization in time series forecasting.
2. The presentation of related work and contributions and the methodology part are **clearly written** and easy to follow.
3. The main forecasting experiments are extended with setups **incorporating missing values and different train portions**, showing some interesting ways of testing models’ generalizability and robustness in forecasting beyond standard experimental setups.

The following weaknesses associated with this work can be easily identified:
- **Incremental Methodological Contribution:** **[W1]** The technical contribution of this work remains limited to extending DeepTime with a regularization term designed for part of ridge regression. It is unclear whether this term could be generally used for other time series forecasting architectures (mainly time-index). **[W2]** Additionally, since ridge regression is a widely used technique in machine learning, the related work section should include references to studies proposing regularization techniques (for standardizing and balancing the correlation of the basis elements or beyond).
- **Issues with Experimental Evaluation:** **[W3]** Several more diverse historical-value state-of-the-art models for time series forecasting could be added as baselines in performance comparisons [1,2,3]. **[W4]** In the experiment with missing points, it is interesting that historical-value models are also evaluated in a fair way (using some simple interpolation method for missing points). **[W5]** Additionally, several methods have been proposed for modeling irregularly sampled / with missing values time series, such as Neural ODEs, RNN-based, and others [4,5], that could potentially outcompete most standard models in the missing points setup. **[W6]** There is also a discrepancy in the number of runs used for PatchTST and the DeepRRTime. The authors mention performing 10 runs for DeepRRTime, and the models they need to re-evaluate. However, the obtained results for PatchTST (for all datasets except the Exchange dataset) from the original paper are averaged over 5 runs.
- **Studies on Computational Cost/Visualizations:** **[W7]** The authors mention that “time-index models are generally less computationally expensive, especially at inference time”, but if not mistaken, this is not accompanied by any time/memory cost comparison between the considered models. **[W8]** Plots on predictions are also missing, which could bring some light to the cases the the regularized DeepTime surpasses the baselines.

1. Liu, Y., Hu, T., Zhang, H., Wu, H., Wang, S., Ma, L., & Long, M. (2023). itransformer: Inverted transformers are effective for time series forecasting. arXiv preprint arXiv:2310.06625.
2. Zhang, Y., & Yan, J. (2023, May). Crossformer: Transformer utilizing cross-dimension dependency for multivariate time series forecasting. In The eleventh international conference on learning representations.
3. Wu, H., Hu, T., Liu, Y., Zhou, H., Wang, J., & Long, M. (2022). Timesnet: Temporal 2d-variation modeling for general time series analysis. arXiv preprint arXiv:2210.02186.
4. Rubanova, Y., Chen, R. T., & Duvenaud, D. K. (2019). Latent ordinary differential equations for irregularly-sampled time series. Advances in neural information processing systems, 32.
5. Yalavarthi, V. K., Madhusudhanan, K., Scholz, R., Ahmed, N., Burchert, J., Jawed, S., ... & Schmidt-Thieme, L. (2024, March). GraFITi: Graphs for Forecasting Irregularly Sampled Time Series. In Proceedings of the AAAI Conference on Artificial Intelligence (Vol. 38, No. 15, pp. 16255-16263).

---

> ### Author Response · Authors · 2024-11-15
> **Review Response**
>
> We thank the reviewer for their time in reviewing our submission, appreciating the strengths and providing insightful recommendations to improve the paper. We address the weaknesses and answer your questions in the following:
>
> **[W1] Incremental methodological contribution**
>
> Our key insight is that the multicollinearity in DeepTime's feature space can lead to suboptimal learning and hence forecasting. Krikheli & Leshem (2021) theoretically analyse the sample-complexity (i.e., minimum number of samples for a desired test-error) of linear-regression problems with correlated features and suggest that it is easier to learn linear models on less mutually correlated features. Since we apply the linear-regression upon the meta-learner’s feature-space, we explore the possibility of directly controlling the linear-correlations and propose the covariance-regularization loss-term to improve the forecast accuracy. To the best of our knowledge, this is a novel application of covariance regularization.
>
> We acknowledge the reviewer’s concern regarding the generalizability of our regularization method beyond the DeepTime model. To address this, we explored applying our regularizer to the TimeFlow framework, the only other deep time-index model currently in the literature to our knowledge. However, as detailed in Appendix C, we encountered methodological limitations when testing TimeFlow, resulting in irreproducible outcomes. Specifically, with our evaluation methodology, TimeFlow’s performance was not competitive with other state-of-the-art models, prompting us to discontinue this line of investigation.
>
> Our preliminary experiments, however, indicate that our regularizer can confer benefits to TimeFlow and has a similar effect of either improving the performance or not impacting it significantly. To incorporate the regularization, we compute the covariance-regularization loss on the basis representations and optimize this when adapting the code to a forecasting window — we refer to this model as TimeRRFlow. The table below shows both of these results illustrated on the ECL and Traffic datasets. For these results, we used 3 seeds due to the longer running time of the TimeFlow model.
>
> |  |  | TimeFlow |  | TimeRRFlow |  |
> |---|---|---|---|---|---|
> |  | H | MAE | MSE | MAE | MSE |
> | ECL | 96 | 0.2401±0.0047 | 0.1408±0.0022 | **0.2365**±0.0018 | **0.1395**±0.0014 |
> | ECL | 192 | 0.2506±0.0006 | 0.1547±0.0012 | **0.2494**±0.0008 | **0.1542**±0.0008 |
> | ECL | 336 | **0.2677**±0.0015 | **0.1699**±0.0008 | 0.2690±0.0019 | 0.1700±0.0018 |
> | ECL | 720 | 0.2997±0.0028 | 0.2027±0.0016 | **0.2986**±0.0007 | **0.2024**±0.0011 |
> | Traffic | 96 | 0.2869±0.0075 | 2.6213±1.2557 | **0.2809**±0.0065 | **1.3516**±0.5771 |
>
> Additionally, we recognize that our regularization method opens promising avenues for future work. We have created a dedicated section in the paper to discuss the following extensions:
> * **Extension to directly control the covariance matrix eigenvalues.** Future work could explore advanced regularization methods to control eigenvalues more explicitly, building on the demonstrated effectiveness of our covariance regularization technique.
> * **Non-linear generalization.** Motivated by the benefits of covariance regularization shown here, future research can extend DeepTime with non-linear base learners (e.g., replacing linear-regression with a more expressive MLP), introducing regularization approaches to manage non-linear correlations.
> * **Extensions to other applications.** Since our method is not specific to time-series forecasting, it has the potential to benefit other applications involving linear base learners within a meta-learner framework (e.g., [1]).
>
> We hope that these additions clarify the novelty, broader applicability and future directions of our work.
>
> [1] Bertinetto et al. Meta-learning with differentiable closed-form solvers. ICLR 2019.
> ***

---

> > ### Author Response · Authors · 2024-11-15
> > **Review Response (contd)**
> >
> > **[W2] Related work on ridge regression**
> >
> > We thank the reviewer for highlighting this connection. Indeed, it is widely discussed that multicollinear data can challenge linear regression methods. Various approaches (e.g., [1, 2, 3, 4])  have been proposed for improving the stability of regression on such data, including additional regularization methods or techniques involving dimensional reduction. However, this entire family of methods has in common that the variates upon which regression is to be conducted are taken as fixed inputs given to the practitioner.
> >
> > In contrast, the DeepTime architecture presents a unique opportunity because the variates used by regression are themselves learned objects. The purpose of our regularizer is to alter the learning dynamics to encourage the variates to avoid collinearity. Evidently, the INR backbone has substantially more capacity than necessary, so that this additional constraint can be added without hurting the model’s expressivity.
> >
> > Because methods for improving the condition of ridge regression are, as a rule, working under the assumption that correlated variates have been provided externally, we believe our approach is fundamentally distinct from those methods. To clarify the difference, we have added the following statement to Sec. 4 of the paper:
> >
> > > We first note the sharp contrast between our method and traditional regularization techniques to improve the conditioning of linear regression on correlated variates, e.g., \cite{adnan2006comparative, paul2006multicollinearity, herawati2018regularized, chan2022mitigating}. As a rule, such methods assume that the correlated variates are specified externally to the problem, and then seek to process them one way or another to minimize the impact of the collinearity on the regression process. In contrast, the INR representation in DeepTime is entirely learned; there are no externally specified collinear variates to disentangle. Our method exploits the unique freedom in this setting to construct these variates specifically such that they lead to a well-conditioned regression problem.
> >
> > [1] Herawati et al. Regularized multiple regression methods to deal with severe multicollinearity. International Journal of Statistics and Applications, 2018.
> >
> > [2] Chan et al. Mitigating the multicollinearity problem and its machine learning approach: a review. Mathematics, 2022.
> >
> > [3] Adnan and Ahmad. A comparative study on some methods for handling multicollinearity problems. MATEMATIKA, 2006
> >
> > [4] Ranjit Kumar Paul. Multicollinearity: Causes, effects and remedies. IASRI, 2006.
> > ***
> > **[W3] Additional historical-value baselines**
> >
> > We thank the reviewer for providing additional references and have added Table 5b, with CrossFormer, TimesNet, and iTransformer baselines, in addition to Table 5a already providing a comparison with four other historical-value methods, including PatchTST, NS Transformer, N-HiTS, ETSformer, FEDformer. As this comparison shows, our method, DeepRRTime, outperforms all three new baselines on 30 out of 40 metrics and ranks second on the remaining 10 metrics. We also note that another historical-value baseline, PatchTST, outperforms all three new baselines on 31 out of 40 metrics.
> > ***
> > **[W4] Evaluation of historical-value methods with missing lookback values**
> >
> > We thank the reviewer for this suggestion. We evaluated PatchTST on the ETTm2 dataset with missing values in the lookback window and reported the results in Table 12. We explored two approaches to handle missing values: (i) replacing missing values with zeros and (ii) applying linear interpolation. The results indicate that DeepRRTime demonstrates significantly greater robustness to missing values compared to PatchTST under these conditions. While we acknowledge that an imputation strategy tailored to a specific dataset and model could yield improved results, we would like to emphasize that one of key advantages of time-index models is their inherent ability to handle missing values natively.
> >
> > ***

---

> > > ### Author Response · Authors · 2024-11-15
> > > **Review Response (contd)**
> > >
> > > **[W5] Neural-ODE/Graph-NN baselines for forecasting with missing lookback values**
> > >
> > > We thank the reviewer for raising this important point. While a fair comparison between time-index models and other approaches, such as Neural ODEs and models based on GNNs, for handling irregularly sampled time series would indeed be valuable, conducting such a comparison would require significant research effort beyond the scope of this paper. In this work, we demonstrate that DeepRRTime, enhanced by our regularization technique, shows substantial robustness to missing values, positioning it as a viable option for applications with irregular time series. The capability of deep time-index models to handle missing values was previously highlighted in the original DeepTime paper, but our results demonstrate that DeepTime can not in fact operate reliably in this setting without our regularizer. Additionally, as our evaluation focuses on long-term forecasting, it differs from prior evaluations of the other deep time-series models capable of handling missing values (e..g, Neural ODEs, GNNs), which have mainly been evaluated on shorter horizons (to our knowledge). We view this as a promising direction for future research, aiming to understand the comparative strengths of time-index models like DeepRRTime in handling missing values relative to other modeling frameworks.
> > > ***
> > > **[W6] Inconsistent number of random seeds in the evaluation**
> > >
> > > We would like to clarify that the results for PatchTST in our paper were taken directly from Table 3 in Nie et al., which are actually based on only a single seed (2021). In their work, PatchTST’s performance variability was assessed using five different seeds, but only to calculate the standard deviation reported in their Table 14. Nevertheless, PatchTST has shown robustness to the choice of random seed; for example, as shown in our Table 12, PatchTST’s performance under a regular forecasting setup remains consistent when evaluated with 10 different random seeds, producing results very close to those obtained with seed 2021.
> > >
> > > In our work, we consistently evaluated our method using 10 different random seeds to obtain more statistically reliable observations. We generally feel that variability is not adequately resolved using extremely small sample sizes (e.g., 5). We used 10 samples as a compromise between our computational budget and our desire for better statistical significance. In a later rebuttal below, we have further confirmed the statistical robustness of our findings by including p-value analyses based on Welch’s t-test.
> > >
> > > ***
> > > **[W7] Computational and memory cost**
> > > We thank the reviewer for this suggestion and have added Table 11 where we provide a comparison between PatchTST and DeepRRTime in terms of peak memory usage and wall-clock time per epoch during both training and inference, using a single NVIDIA Tesla V GPU (16GB). Our results show that DeepRRTime is consistently faster than PatchTST in both training and evaluation. While DeepRRTime is generally more memory-efficient, some exceptions were observed (e.g., Exchange/720). Notably, DeepRRTime did not encounter any out-of-memory (OOM) issues across all forecast horizons on this GPU, whereas PatchTST experienced OOM errors for all horizons on the Traffic dataset. These findings highlight the computational efficiency of our model DeepRRTime compared to historical-value models such as PatchTST.
> > >
> > > ***
> > >
> > > **[W8] Visualisations of model predictions**
> > >
> > > We added visualizations comparing the forecasts of DeepTime, DeepRRTime and PatchTST in Figure 9 (in appendix) on the Exchange dataset. We selected five random examples from the test set (corresponding to the five columns of the figure), and visualized the predictions separately for each column of the dataset (corresponding to the eight rows of the figure).  Interpreting these plots together with the results in Table 1, we can make the following observations: i) PatchTST clearly underperforms the time-index models on this challenging dataset both qualitatively and quantitatively; and ii) the qualitative effect of the regularization term on the DeepRRTime forecasts as compared to DeepTime is also clearly visible: for instance the DeepTime forecasts exhibit far more variance than those of DeepRRTime.
> > >
> > > ***
> > > We hope this response addresses all your outstanding concerns, and we look forward to resolving any additional questions during the remaining discussion period.

---

### Author Response · Authors · 2024-11-15
**Summary of Review Responses**

We thank all the reviewers for taking the time to review our submission and for providing insightful suggestions to improve our work. We are pleased that all four reviewers appreciated the clarity of our presentation, the theoretical motivations, and the extensive experiments. Before addressing each reviewer's comments individually, we would like to restate the goal of our work and outline our claims and evidence.

**Goal.** Our goal is to advance the state-of-the-art in deep time-index models, a relatively new class of deep time-series forecasting models that differs from the prevailing paradigm of deep historical-value models in several important ways. To achieve this, we propose a straightforward, theoretically grounded regularization method designed to enhance the performance of the current leading time-index model, DeepTime.

**Claims and Evidence.** We present the following claims in this work, supported by theoretical and empirical evidence where applicable:
* Covariance regularization enhances the robustness of linear regression.
   - Evidence: Theoretical discussion in section 4 and Appendix A.

* DeepRRTime improves upon (or matches) DeepTime for regular forecasting.
   - Evidence: Tables 1, 3 and 6.

* DeepRRTime improves upon (or matches) DeepTime for forecasting based on missing values.
   - Evidence: Tables 2, 3 and 9. Figures 2 and 8.

* DeepRRTime consistently improves upon DeepTime when training with smaller datasets.
   - Evidence: Table 3 and  4 and Figure 3.

* DeepRRTime improves upon DeepTime when forecasting at a higher frequency at test time as compared to train.
   - Evidence: Figure 5 and 7.

* DeepRRTime is more compute-efficient than PatchTST.
   - Evidence: Table 11.

Based on the reviews, we collected additional results and presented them in this rebuttal:
* [kRWA, ZUzk] Computational and memory cost comparison for DeepRRTime and PatchTST.
* [kRWA, V2ZN] Added more baselines: CrossFormer, TimesNet, iTransformer, and Repeat baseline.
* [ZUzk, kRWA, V2ZN] Evaluation of PatchTST and TimeFlow with missing lookback values.
* [ZUzk] Discussion about statistical significance of the presented results.
* [ZUzk] Extended comparison of DeepTime and DeepRRTime with missing lookback values.
* [V2ZN, kRWA] Visualisations of DeepTime, DeepRRTime and PatchTST forecasts.

Please note that all the changes in the submission are highlighted with red color.

---

### Decision · Action_Editor_GGzA · 2025-01-02

**Recommendation:** Accept with minor revision

**Comment:**

While 2 reviewers leaning to Accept and 2 leaning to Reject, all reviewers answered to both questions required by TMLR policy for acceptance with "yes". After reading through the latest revision of the manuscript, the reviews and rebuttal, in my opinion the paper meets the criteria for acceptance. I find the novelty quite limited, but the theoretical insight worthy of the attention of the readers of TMLR.

I would ask the authors to extend the text by specifically acknowledging that there are other type of methods for sporadically observed data which was not tested in this work, like Neural ODEs. This was mentioned by the reviewers, and the authors decided that this comparison would be outside of the scope of the present work. I can live with this decision but I find it essential to mention in the manuscript the existence of these methods.

**Audience:**

The audience would be interested in the findings according to all reviewers.

**Claims And Evidence:**

The claims supported by evidence according to all reviewers.

---

> ### Author Response · Authors · 2025-01-30
> **Camera-ready revision**
>
> We thank the Action Editor and the reviewers for their time in reviewing our submission and providing invaluable suggestions towards improving our paper. The camera-ready revision includes the following updates to our earlier revision:
> * Figure 8 is updated with remaining datasets/horizons/seeds.
> * Acknowledged alternatives for handling irregular time-series.